# Nutrient levels control root growth responses to high ambient temperature in plants

Sanghwa Lee [1], Julia Showalter[1], Ling Zhang[1], Gaëlle Cassin-Ross [2], Hatem Rouached [2] & Wolfgang Busch [1] ✉

Global warming will lead to significantly increased temperatures on earth. Plants respond to high ambient temperature with altered developmental and growth programs, termed thermomorphogenesis. Here we show that thermomorphogenesis is conserved in Arabidopsis, soybean, and rice and that it is linked to a decrease in the levels of the two macronutrients nitrogen and phosphorus. We also find that low external levels of these nutrients abolish root growth responses to high ambient temperature. We show that in Arabidopsis, this suppression is due to the function of the transcription factor *ELONGATED HYPOCOTYL 5* (*HY5*) and its transcriptional regulation of the transceptor *NITRATE TRANSPORTER 1.1* (*NRT1.1*). Soybean and Rice homologs of these genes are expressed consistently with a conserved role in regulating temperature responses in a nitrogen and phosphorus level dependent manner. Overall, our data show that root thermomorphogenesis is a conserved feature in species of the two major groups of angiosperms, monocots and dicots, that it leads to a reduction of nutrient levels in the plant, and that it is dependent on environmental nitrogen and phosphorus supply, a regulatory process mediated by the HY5-NRT1.1 module.

The recent rise in global temperature is largely due to human activities and is predicted to continue. Most optimistic scenarios predict a 1.5 °C increase of global average temperatures by mid-century, while the middle of the road scenarios predict 2.7 °C increase by the end of the century[1]. Temperature profoundly affects biological systems due to its effect on the free energy for biochemical reactions according to the basic principles of thermodynamics[2,3]. Plants do not regulate their internal temperature and due to their sessile nature, they are very sensitive to climate change[4]. Plants respond to high ambient temperature with a developmental program termed thermomorphogenesis. The hall-mark phenotypes of thermomorphogenesis are elongated tissues including hypocotyl, petiole and root, hyponastic growth, stomatal development, and early flowering[5–8].

There are two transcription factors serving as central hub in thermomorphogenesis, PHYTOCHROME INTERACTING FACTOR 4

(PIF4) and ELONGATED HYPOCOTYL 5 (HY5) both of which were originally identified as light signaling components[6,9–11]. PIF4 has a major role in the thermomorphogenesis of the shoot, which also involves other PIFs such as PIF1, 3, 5, and 7[10,12–16]. HY5 plays a major role in root thermomorphogenesis, which regulates primary root length at the early seedling stage[5,6].

Another key factor for growth and development is nutrient availability. There is a strong interaction of nutrient availability and temperature for determining growth[17]. Conversely, nutrient content is affected by elevated temperature and increased CO$_2$ levels[18–20]. Nitrogen (N), Phosphorus (P), and Potassium (K) are three macronutrients which are commonly used for agricultural fertilizer and to enhance growth. Due to the importance of macronutrient uptake, nutrient signaling has been widely studied. One of the most well-studied genes in nitrogen uptake is *NITRATE TRANSPORTER 1.1* (*NRT1.1*), which

[1]Plant Molecular and Cellular Biology Laboratory, Salk Institute for Biological Studies, 10010 N Torrey Pines Rd, La Jolla, CA 92037, USA. [2]Department of Plant, Soil, and Microbial Sciences, Michigan State University, East Lansing, MI 48823, USA. ✉e-mail: wbusch@salk.edu

encodes for a dual-affinity nitrate transporter[21] and nitrate sensor[22–25]. Furthermore, *OsNRT1.1B*, which is a functional homolog of *AtNRT1.1* in rice, has been showed to integrate N and P signaling[26], suggesting that the role of NRT1.1 as a master regulator of N and P might be conserved across the plant kingdom.

Here, we show that shoot and root thermomorphogenesis are conserved among Arabidopsis (*Arabidopsis thaliana*), soybean (*Glycine max*), and rice (*Oryza sativa*). We find that this is linked to decreased N and P levels in plant tissues at higher temperatures. Conversely, low levels of N and P in the growth medium abolished thermomorphogenesis in Arabidopsis. We found that a module constituted by the thermomorphogenesis key regulator *HY5* and the nitrogen transceptor *NRT1.1* is in involved in this regulation process.

## Result

### Plants show conserved shoot and root thermomorphogenesis that goes along with decreased N and P levels in plant tissues

Root thermomorphogenesis studies have been largely restricted to Arabidopsis. We therefore wanted to compare this to the response in other species. For this, we grew Arabidopsis, rice, and soybean seedlings at ambient and elevated temperatures. For Arabidopsis, Col-0 wild-type seedlings were grown at either 21 °C or 28 °C for 5 days after 4 days of germination at 21°C (Fig. 1a, b). Soybean (Williams 82 variety) and rice (Kitaake ecotype) were grown at either 28 °C or 33 °C for 1 or 2 weeks for soybean and rice, respectively, after 7 days of germination at 28 °C (Fig. 1c-f). Similar to the reported Arabidopsis shoot and root thermomorphogenesis[6,10] (Fig.1 a, b), both soybean and rice showed longer shoots and roots at higher temperatures, indicating that plants have a conserved elongation mechanism at higher temperatures.

Previous studies had exposed temperature-dependent gene expression changes of gene clusters related to nitrogen and other nutrient-related processes[5,6] (Supplementary Data 1) Furthermore, HY5 has been found to regulate nitrogen[27,28] and iron signaling[29,30]. We therefore hypothesized that nutrient composition or uptake could be changed at higher temperature. To test this hypothesis, we analyzed the nutrient composition of shoots of 4 week-grown plants of Arabidopsis, rice, and soybean (Fig. 1g–i, Supplementary Fig. 1). We also included Arabidopsis *hy5-215*[11] and *pifQ*[31] mutants in this analysis. Interestingly, levels of N and P were decreased in Arabidopsis, soybean, and rice at higher temperatures (Fig. 1g–i), while other nutrients showed less consistent patterns across the species at higher temperatures (Supplementary Fig. 1). As previous transcriptome experiments in *hy5* mutants had shown nitrogen related processes to be altered[5,6] (Supplementary Data 1), we hypothesized that the observed temperature-dependent changes in N-levels were mediated by *HY5*. However, N-levels in the *hy5-215* mutant were not statistically different from Col-0 (Fig. 1g). This might suggest that while *HY5* is involved in transcriptional changes of genes involved in nitrogen-related processes, it isn't required for the changes in N levels that are observed at high ambient temperature. Because of the prominence of P-level changes that we had observed at higher ambient temperatures, we also measured P in these genotypes. In contrast to N, the *hy5-215* mutant plants showed a significant, opposite change of P level changes in response to higher ambient temperature compared to the WT (Fig. 1i). *pifQ* was similar to Col-0 at high ambient temperature (Fig. 1g–i) indicating that PIFs are not required for this. Overall, our results showed that *HY5* is involved in the alteration of P levels of mature plant shoots that were observed at high ambient temperatures. As HY5's impact in thermomorphogenesis can be observed in young seedlings, we wanted to investigate whether similar patterns of nutrient changes among genotypes and temperatures were detectable in the roots of young seedlings. For this, we measured nutrient contents of 9-day-old seedling roots from Arabidopsis, soybean, and rice. However, in these samples we didn't detect the same trends as observed in older plant roots (Supplementary Fig 2). For example, we observed decreased N-levels and increased P-levels in rice, while Arabidopsis and soybean did not show any

differences at higher temperature. These data suggest that changes of nutrients accumulate over time in plants and then cumulatively affect nutrient contents in the shoot part of more mature plants. Alternatively, it is also possible that this is a time-dependent regulatory process that starts later than 9 days after germination. Taken together our data show that the levels of N and P in plants are regulated in response to elevated temperature, and that in Arabidopsis HY5 is required for the regulation of temperature-dependent P level changes.

### HY5 integrates temperature and N–P signaling and directly represses *NRT1.1* transcription

Since we found *HY5* to be involved in temperature-dependent P-level changes in Arabidopsis and at least at the transcriptional level affected genes involved in nitrogen-related processes at high ambient temperature, we searched for a target downstream of *HY5* that could explain its function. HY5 is a bZIP protein transcription factor, which binds to several DNA sequence motifs including G-box (CACGTG) and CACGT motifs[32,33]. Published ChIP-seq data showed that HY5 binds to the promoter region of genes that are involved in nutrient-related responses such as those that we had identified using our RNAseq to be related to nitrogen and organic acids[6,32,33]; Supplementary Data 1). We identified genes that were bound by HY5 according to the ChIP-seq data[32] and that are in the N–P signaling pathway. These genes included N signaling pathway genes such as *NIGT1.1, HHO2, NLP7, NRT1.1, NRT1.5, NRT2.1, NIA1*, and *LBD37*[23,34–39]. Interestingly, promoter regions of P signaling pathway genes including *PHO1, PHT1;8*, and *IPS1* did not show high enrichment in the HY5 ChIP-seq data (Supplementary Fig. 3). To examine whether HY5 directly binds to the promoter of the genes from the ChIP-seq analysis in our growth conditions (the published ChIP-seq data were obtained under different light conditions), we performed Chromatin Immunoprecipitation qPCR (ChIP-qPCR) using 4 days 21 °C grown *pHY5:HY5-GFP* whole seedlings with an additional 5 days growth at either 21 °C or 28 °C (Fig. 2a, b, supplementary Fig. 4a, b). Interestingly, we detected enrichment at high ambient temperature for only a subset of target promoters, including *NIGT1.1, HHO2, NLP7*, and *NRT1.1*. This indicates that HY5 directly binds to N signaling genes, with NRT1.1 also being involved in the integration of N and P signaling[26,40]. This might indicate that HY5 might directly regulate expression in a temperature dependent manner and thereby exert influence on N–P signaling. To test whether the transcription levels of those genes are altered at high ambient temperature, we performed qPCR of root and shoot tissues (Fig. 2c, supplementary Fig. 4c). Consistent with our ChIP-qPCR data, temperature-dependent HY5 enriched target genes such as *NIGT1.1, HHO2, NLP7*, and *NRT1.1* were significantly downregulated while the genes for which we hadn't found ChIP-qPCR enrichment, such as *NRT1.5, NRT2.1, NIA1, and LBD37* were not altered at high ambient temperatures. Among the genes that were bound by *HY5* and downregulated, *NRT1.1* stood out as its transcript levels were strongly downregulated at high ambient temperature in the roots of Col-0, but this change of its expression level was abolished in the roots of *hy5-215* mutant plants. This was different in the shoot, as downregulation of *NRT1.1* upon high ambient temperatures and its dependency on *HY5* was much less pronounced there. Overall, this suggested that the transcript level of *NRT1.1* is tightly and directly regulated by HY5 in the root in a temperature-dependent manner. To further test whether HY5 directly binds to the promoter region of *NRT1.1*, we performed an Electrophoretic Mobility Shift Assay (EMSA) (Fig. 2d). Consistent with our hypothesis that HY5 binds to the *NRT1.1* promoter region, HY5 was able to bind to the G-box motif in the *NRT1.1* promoter region. Furthermore, we tested whether HY5 acts as a transcription repressor of *NRT1.1* transcription by using a dual luciferase assay in *Nicotiana benthamiana* (Fig. 2e). Consistent with our hypothesis that HY5 directly binds to the *NRT1.1* promoter and represses its transcription, transcription of the reporter was reduced in the presence of HY5, suggesting that HY5 represses *NRT1.1*

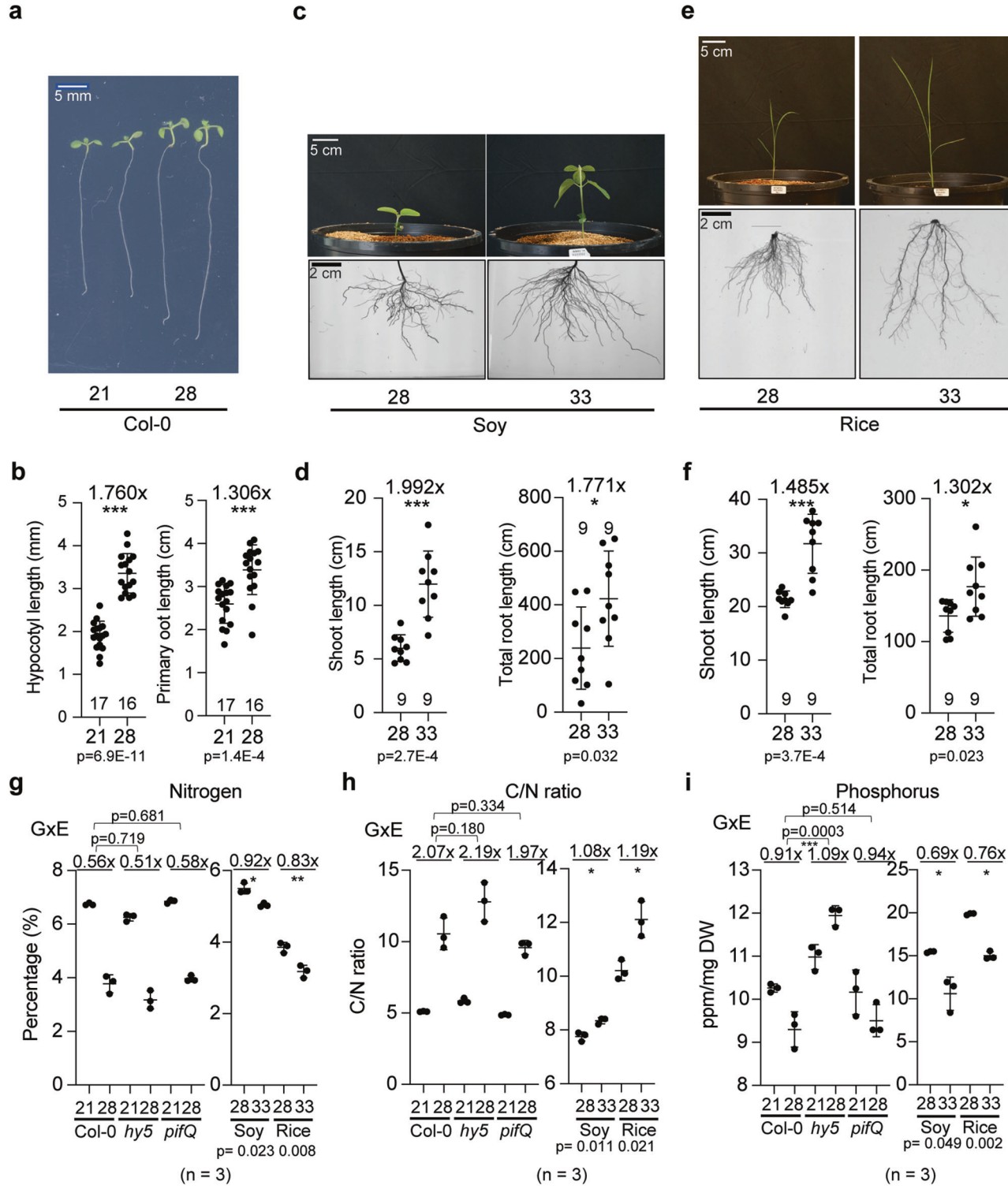

**Fig. 1 | Plant developmental responses to high ambient temperatures are conserved and are linked to altered nutrient levels. a–f** Phenotypes of Arabidopsis (Col-0; **a**, **b**), soybean (Williams82; **c**, **d**), and rice (Kitaake; **e**, **f**) at normal and higher temperatures. Arabidopsis seedlings were grown for 4 days on 1/2 ms plates at 21 °C and then kept either at 21 °C or 28 °C for 5 additional days. Rice and soybean seedlings were grown for 1 week at 28 °C and then either kept at 28 °C or 33 °C for additional 2 weeks for rice and 1 week for soybean, respectively. Scatter dot plot shows average difference in primary root length for Arabidopsis, and total root length for soybean and rice, and the number of plants. p-Value from one-sided Student's *t* test. **g, h** Nitrogen (**g**) and C/N ratio (**h**) in Arabidopsis shoots (Col-0,

*hy5-215*, and *pifQ*), soybean shoots, and rice shoots using CN analysis. **i** Phosphorus in Arabidopsis shoots, Soybean shoots, and rice shoots using MP-AES. For (**g–i**), *n* = 3 biologically independent samples were used. *p*-Values for the corresponding GxE interactions determined through ANOVA are shown on top of each graph. Asterisks indicate statistically significant difference either 2-way ANOVA or one-sided Student's *t* test; *$p < 0.05$, **$p < 0.01$, ***$p < 0.001$, and ****$p < 0.0001$. Average fold difference of each group is indicated in the top region of the plot. Shoot parts from 4-week-old plants from Arabidopsis, soybean, and rice plants were used for the nutrient analyses. Plots indicate mean (horizontal line) and standard deviation (error bars).

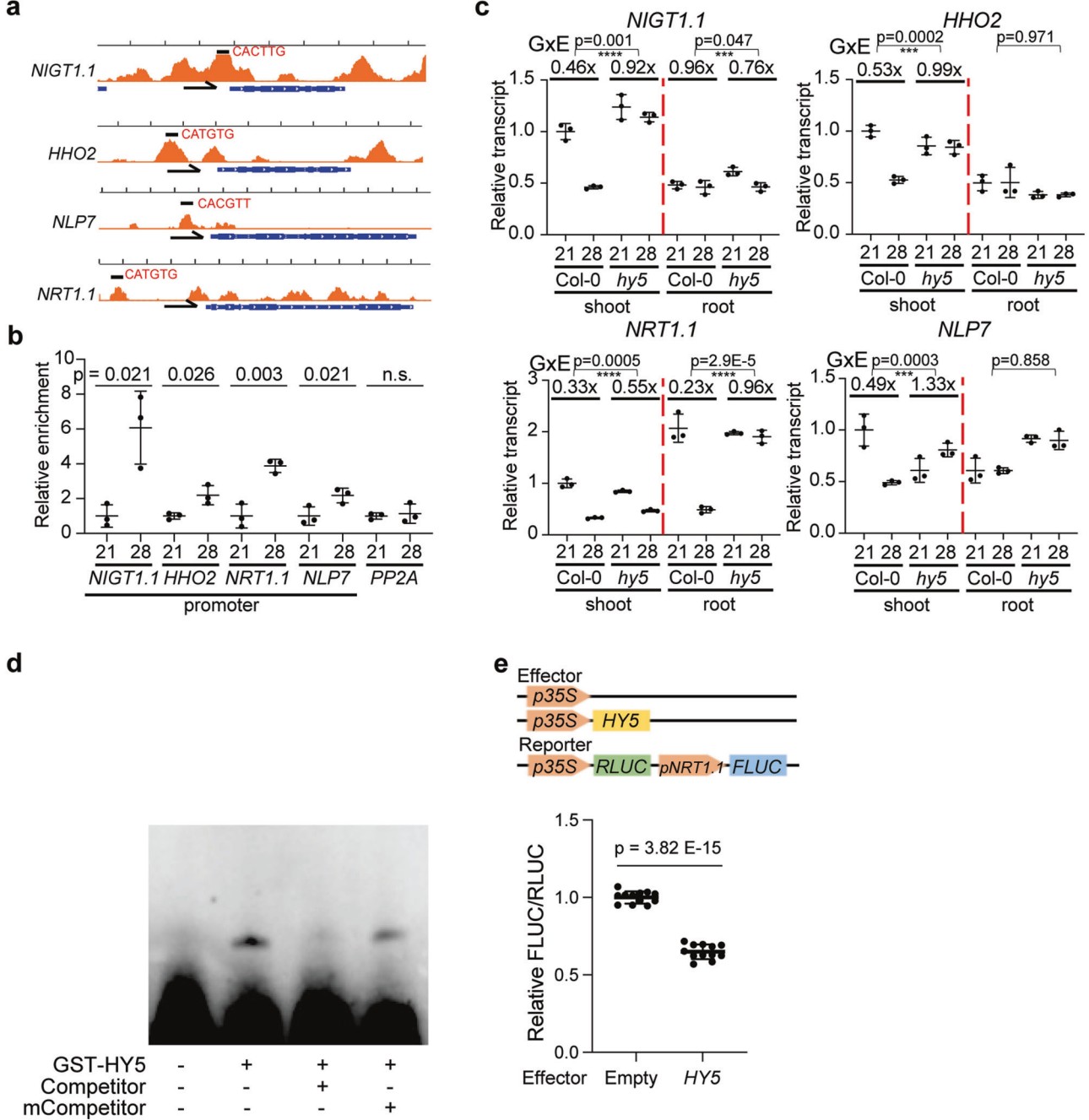

**Fig. 2 | HY5 integrates temperature and N–P signaling and directly represses *NRT1.1* transcription. a** IGV image of HY5 ChIP-seq data from Burko et al.[32] of selected N–P signaling genes with transcription direction and binding motif. **b** Scatter dot plot of ChIP-qPCR results at normal and high ambient temperature of promoter regions of five different genes. *p*-Values for the two-sided Student's *t* test. **c** Scatter dot plot of qPCR results at normal and high ambient temperature of four different genes using Col-0 and *hy5* shoot and root samples of seedlings. Relative transcript level was normalized using *PP2A* as a control and to the expression levels in the shoot. For (**b** and **c**), *n* = 3 biologically independent samples were used. p-Values for the corresponding GxE interactions determined through ANOVA are shown on top of each graph. Asterisks indicate statistically significant

difference either 2-way ANOVA; *$p < 0.05$, **$p < 0.01$, ***$p < 0.001$, and ****$p < 0.0001$. **d** Electrophoretic Mobility Shift Assay (EMSA) showing GST-HY5 binds to G-box motif of NRT1.1 promoter region. G-box motif containing nucleotides were biotin labeled. Competitor is the same sequence but biotin unlabeled. mCompetitor is mutated version of G-box motif (CACATG to CCCATG) without biotin label. Two independent experiments were repeated with similar results. **e** dual luciferase assay using *Nicotiana benthamiana*. (upper) Effector and reporter constructs are described. (lower) Relative FLUC/RLUC showing HY5 as a transcriptional repressor. *n* = 12 biologically independent samples were used. *p*-Value from two-sided Student's *t* test. Plots from (**b**, **c**, and **e**) indicate mean (horizontal line) and standard deviation (error bars).

transcription through direct binding to its promoter region. Taken together, our data strongly suggest that HY5 directly binds to the promoter region of *NRT1.1* and represses its transcription at high ambient temperature in the root. However, as we performed ChIP-qPCR in whole seedlings, EMSA in vitro, and the dual luciferase assay in *Nicotiana benthamiana*, it can't be fully excluded that the direct

binding of HY5 to the *NRT1.1* promoter, for some unknown reason (even through HY5 and NRT1.1 are expressed in the root), might not occur in the root.

In contrast to NRT1.1, further investigation of other candidate genes did not provide strong support of their direct regulation by HY5. While the transcript level of *NIA1* was altered in *hy5-215* mutant, there

was no indication of a change in HY5 binding at high ambient temperature according to our ChIP-qPCR (Supplementary Fig. 4a–c), suggesting that *NIA1* transcript level is altered indirectly. Furthermore, the transcript level of *NRT1.5* decreased at high ambient temperature both in Col-0 and *hy5-215* mutant (Supplementary Fig. 4a-c), indicating that other components might be responsible for regulating *NRT1.5* transcript levels at high ambient temperature. Taken together, our data suggest that HY5 regulates root thermomophogenesis, transcriptional programs relating to nitrogen, and phosphate levels by repressing the N–P signaling genes and by directly regulating key genes such as *NRT1.1*.

### Root thermomorphogenesis depends on external N–P levels and phosphorylation but shoot to root mobility of HY5 is not required for this

Because *HY5* is involved in the regulation of genes that play a role in nitrogen and nutrient-related processes and is required for the appropriate *P* level changes in response to high ambient temperature (Supplementary Data 1, Fig. 1g–i), we hypothesized that *HY5* levels might be affected by N–P deficient conditions at high ambient temperature. To test this hypothesis, Col-0 plants were grown for 5 days at either 21 °C or 28 °C after 4 days for germination in 21 °C in three different media conditions: 1/2MS (N:11400 μM, P: 625 μM), mildly nitrogen deficient (N: 550 μM), and mildly phosphorus deficient (P: 100 μM)[41]. *HY5* transcript levels only changed in ½ MS medium in response to high ambient temperature, but not in -N or -P conditions (Fig. 3a). Similar to a short time of exposure to high ambient temperature for 4 h[6], HY5 protein levels were elevated even after a long time of high-temperature exposure for 5 days in nutrient sufficient ½ MS media. Consistent with the transcript levels, HY5 protein levels did not change in N or P deficient media under high temperature (Fig. 3b).

We then set out to test whether the altered HY5 level affected root thermomorphogenesis in these three different conditions. As it had been shown that SPA mediated HY5 phosphorylation is crucial for root thermomorphogenesis[6], we not only utilized the *hy5-215* mutant line but also three transgenic overexpression lines in the *hy5-215* mutant background: *35 S:HY5-GFP*, *35 S:HY5 S36A-GFP*, and *35 S:HY5 S36D-GFP*. In wildtype, root thermomorphogenesis was observed only in ½ MS grown seedlings, which displayed increased HY5 protein levels at high ambient temperature (Fig. 3a, b). Grown on N or P deficient media, wildtype seedlings did not show a longer primary root at high ambient temperature. This was similar to the response of the *hy5-215* mutant, in which there is no HY5 protein produced and that didn't even display increased root length in nutrient sufficient conditions (Fig. 3b, c). HY5 plants overexpressing HY5 or a HY5 phospho-mimic S36D version could not only complement the *hy5-215* mutant phenotype in nutrient sufficient conditions, but in addition led to thermomorphogenesis in -N, and -P media. In contrast, overexpression of the phospho-dead S36A HY5 version did not complement the *hy5-215* mutant phenotype in any of the conditions. This shows that phosphorylation of HY5 is necessary for HY5 activation and stabilization during thermomorphogenesis and HY5 phosphorylation is sufficient for eliciting thermomorphogenesis under otherwise restrictive condition such as low environmental levels of N or P. It also might suggest that SPA dependent phosphorylation is involved in this. Overall, these data suggest that phosphorylation of HY5 is essential for temperature mediated primary root elongation and that the lack of temperature dependent root growth response in N or P deficient medium is due to the lack of increased HY5 levels.

We then tested the effect of excessive amount of N or P on root thermomorphogenesis. For this, Col-0 seedlings were grown in N or P excessive media at high ambient temperature (Supplementary Fig. 5). Col-0 seedlings did not display exaggerated root thermomorphogenesis in N or P excessive media at high ambient temperature when comparing it to 1/2MS conditions, indicating that excessive amounts of

N or P do not affect root elongation at high ambient temperature. Taken together, these data suggest that sufficient amounts of N and P are required for HY5 mediated root thermomorphogenesis.

While it is not yet fully resolved whether HY5 movement from shoot to root is always necessary for root thermomorphogenesis, we tested this with regards to the inhibition of root thermomorphogenesis by low N and P levels. For this, we excised the roots of 4-day-old Col-0 and *hy5-215* seedlings and grew them for additional 5 days in N or P deficient media at high ambient temperature (Fig. 3e). Excised Col-0 roots were able to respond to high ambient temperature in ½ MS but not in N or P deficient media. Excised *hy5-215* roots were not able to respond to the elevated temperature in any of the conditions. Overall, this is supportive of a major root autonomous function of HY5 in root thermomorphogenesis and its alteration by low N and P levels.

### NRT1.1 integrates N and P dependent root thermomorphogenesis

NRT1.1 has been shown to be a key component in nitrate signaling and is regulated by nitrate and phosphate level, as well as by auxin in root development[23,40,42]. NRT1.1 protein level is destabilized during P starvation in Arabidopsis[40] and the OsNRT1.1B-OsSPX4 module has been identified to integrate nitrate and phosphate signaling in rice[26]. As *NRT1.1* transcript levels were decreased at high ambient temperature and were tightly regulated by HY5 in a root specific manner via its direct transcriptional suppression (Fig. 2c-f), we performed qPCR using roots of Col-0 and a HY5 overexpressing line (*35 S:HY5-GFP/hy5-215*) grown on nutrient sufficient or deficient media (Fig. 4a). These plants were grown at either 21 °C or 28 °C for 5 days after they had germinated and grown for 4 days at 21 °C. Seedlings grown on nutrient sufficient medium displayed significantly decreased transcript levels of *NRT1.1* at high ambient temperatures. Seedlings grown on N or P deficient media showed a decreased level of *NRT1.1* transcript at 21 °C (albeit not as low as those grown on at 28 °C and nutrient sufficient medium) and this was not further decreased in high ambient temperature. This might suggest that sufficient amounts of N and P are necessary for the high level of NRT1.1 transcripts at 21 °C and that nutrient deficiency and high ambient temperature independently reduces NRT1.1 transcript levels to the lowest level. Furthermore, the HY5 overexpression line showed decreased transcript levels of *NRT1.1* in all tested nutrient conditions (½ MS, -N, and -P media) at high ambient temperature, which is in line with the thermomorphogenesis phenotypes in these plants (Fig. 3b, c). Overall, these results indicate that while *NRT1.1* transcript levels might be independent from HY5 at 21 °C, HY5 is able to repress *NRT1.1* transcription at high ambient temperature to trigger root thermomorphogenesis.

To investigate these transcriptional changes in higher detail, we analyzed seedlings of the *pNRT1.1:GFP* reporter line[43] grown at 21 °C or 28 °C, to assess tissue specific expression changes in different nutrient conditions using confocal microscopy (Fig. 4b, c). Reflecting the data from the qPCR, the *pNRT1.1:GFP* signal was decreased in the root apex at high ambient temperature in roots grown on nutrient sufficient medium. The GFP signal was much lower in roots grown on N or P deficient media at both temperatures (Fig. 4b, c). Overall, these data suggest that sufficient amounts of N and P are necessary for the suppression of *NRT1.1* transcript level at high ambient temperature.

As *NRT1.1* transcript level is altered in different media and temperature conditions (Fig. 4a–c), we measured total root protein levels of NRT1.1 using a native NRT1.1 antibody (Fig. 4d). Similar to the transcriptional response, NRT1.1 protein levels decreased at high ambient temperature under nutrient sufficient conditions (Fig. 2c, Fig. 4a). This indicates that both transcription and protein level of NRT1.1 are decreased at high ambient temperature. Importantly, the decrease of NRT1.1 protein level observed at high ambient temperature, was less pronounced in -N and -P media (Fig. 4a, d). The HY5 overexpression line showed strongly decreased NRT1.1 protein levels

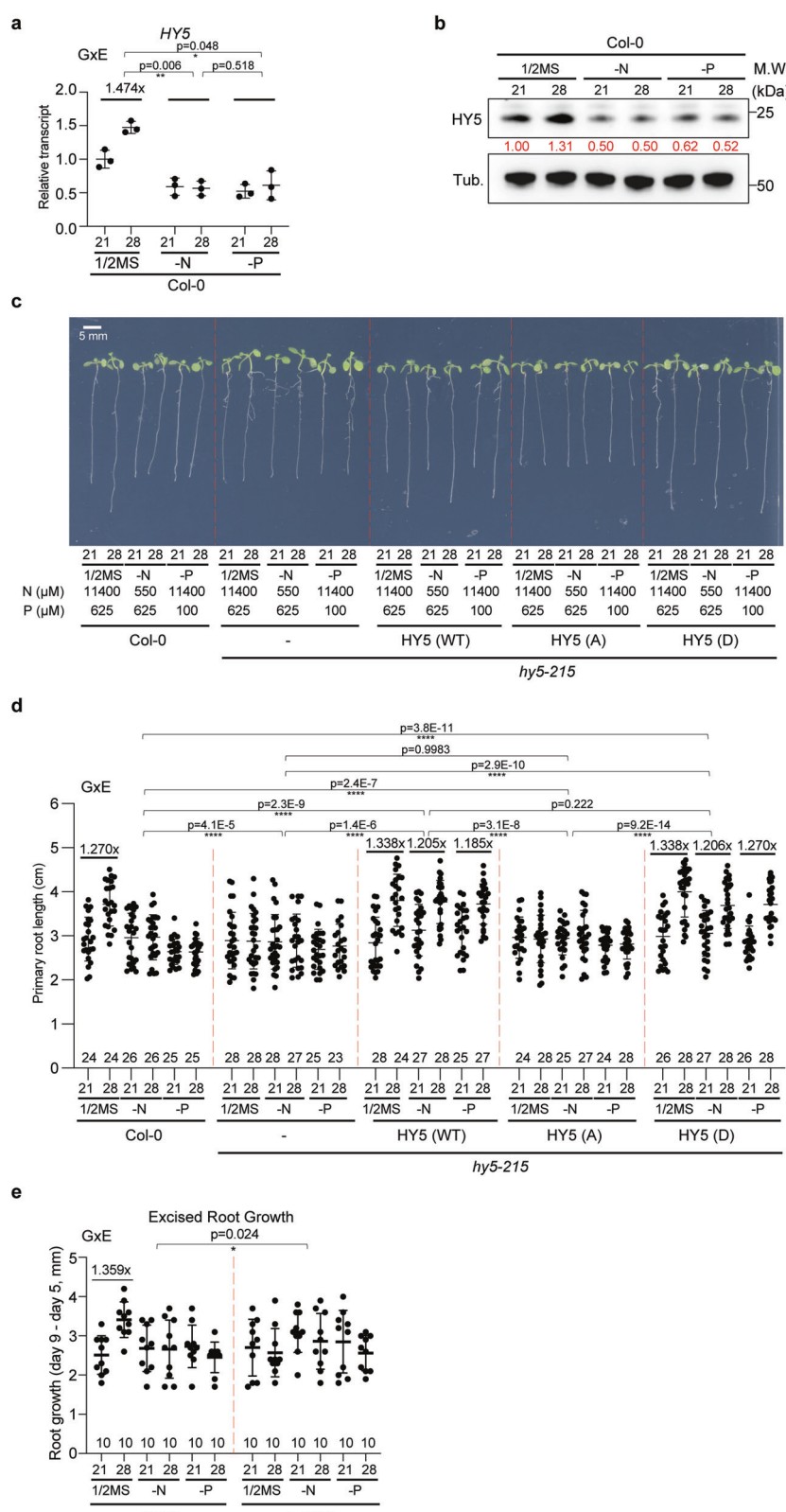

in all nutrient conditions (½ MS, -N, and -P media) at high ambient temperature (Fig. 4d) showing that transcriptional downregulation strongly affects NRT1.1 protein levels at high ambient temperature. To assess whether protein degradation also contributes to the observed decrease of NRT1.1 protein level at high ambient temperatures, we treated the samples with cycloheximide (CHX) and ES9-17 for 4 h, which are protein synthesis and clathrin-mediated endocytosis inhibitors, respectively (Supplementary Fig. 6). We reasoned that any NRT1.1 level change that is observed between CHX treatment and concomitant treatment with CHX and ES9-17 would be indicate of inhibiting NRT1.1 degradation (through inhibiting clathrin-mediated endocytosis). Concomitant treatment with CHX and ES9-17 resulted in increased levels of NRT1.1 in all treatments when compared to CHX treatments alone, indicating that the observed change of NRT1.1

**Fig. 3 | Root thermomorphogenesis depends on external N–P levels and phosphorylation but not shoot to root mobility of HY5 is required for this.** **a, b** HY5 transcript (**a**) and protein (**b**) level grown on different media at control and high ambient temperature. Root samples were analyzed separately. For (**a**), *n* = 3 biologically independent samples were used. Native HY5 antibody was used for Western blot. Red number indicates the relative signal intensity divided by HY5 signal to Tubulin. **c** Phenotypes of Col-0, *hy5-215*, and 3 different forms or HY5 overexpression lines (WT, A: phospho-dead, D: phospho-mimic) in *hy5-215* grown

on different media at control and high ambient temperatures. **d** Scatter dot plot of (**c**). **e** Phenotypes of excised roots grown on different media at control and high ambient temperature with the number of plants indicated. Average fold difference of each group is indicated in the top region of the plot. Scatter dot plots indicate mean (horizontal line) and standard deviation (error bars). *p*-Values for the corresponding GxE interactions determined through ANOVA are shown on top of each graph. Asterisks indicate statistically significant difference either 2-way ANOVA; *$p < 0.05$, **$p < 0.01$, ***$p < 0.001$, and ****$p < 0.0001$.

protein levels at high ambient temperature and in -N and -P conditions are at least partly due to post-translational regulation and degradation. To further corroborate NRT1.1 regulation at the protein level, we examined it in the root apex at cellular resolution. For this, we utilized *pNRT1.1:NRT1.1-GFP* transgenic plants[23] (Fig. 4e, f). Consistent with the data from the western blots, NRT1.1 protein level was decreased in the root apex in ½ MS media at high ambient temperature, while N or P deficient media grown seedlings did not show decreased *NRT1.1-GFP* signal in the root tip (Fig. 4e, f). Overall, the data suggest that changes in NRT1.1 transcript and protein level might be necessary for root thermomorphogenesis.

### The *HY5-NRT1.1* regulatory module is required for the interaction of root thermomorphogenesis and P level

As *NRT1.1* transcript level was tightly regulated by HY5 in the root at high ambient temperatures (Fig. 2c–f), we examined if there is a *HY5* feedback loop from NRT1.1 on *HY5*. *HY5* transcript levels showed similar patterns in the *NRT1.1* loss of function mutant line (*chl1-5*) compared to Col-0 in the root, where *HY5* transcript level is induced at high ambient temperature, and a slightly increased response to high temperature in the shoot (Fig. 5a). This indicated that there is no significant feedback from NRT1.1 on the *HY5* transcript level in the root. To test this at the protein level we performed western blots (Fig. 5b). Interestingly, HY5 protein level was more accumulated in *chl1-5* mutant roots in all our conditions, suggesting that *NRT1.1* might regulate HY5 protein stability directly or indirectly via unknown mechanisms. Furthermore, NRT1.1 protein accumulated to a higher extent in *hy5-215* mutant roots at high ambient temperature, which was consistent with its transcript level (Fig. 2c), indicating that HY5 has an important role to suppress *NRT1.1* transcript level and thus regulates NRT1.1 protein level at high ambient temperature.

The direct binding of HY5 to the *NRT1.1* promoter and the transcriptional regulation of *NRT1.1* expression level by *HY5* (Fig. 2a–e) had indicated that HY5 is upstream of NRT1.1. To genetically test this, we obtained mutant lines of *HY5* (*hy5-215*), *NRT1.1* (*chl1-5*), double mutant (*hy5-215 chl1-5*). We then assessed hypocotyl and primary root length, as well as N–P composition at standard and high ambient temperatures (Fig. 5c–g). As reported previously[5,6], *hy5-215* showed a thermo-insensitive root phenotype. The *chl1-5* mutant showed an increased root elongation phenotype at high ambient temperature compared to that of Col-0, while displaying a similar pattern of shoot thermo-morphogenesis (Fig. 5c, d). Consistent with our data that HY5 represses *NRT1.1* expression in a root specific manner at high ambient temperature (Fig. 2c), *hy5-215 chl1-5* double mutant plants showed similar phenotypes to *hy5-215* in the shoot and *chl1.5* in the root, respectively (Fig. 5c–e). This indicates that *chl1-5* and *hy5-215* are additive towards hypocotyl and primary root length in response to temperature. To further support our model, we also checked whether overexpressed *NRT1.1* could repress root thermomorphogenesis using a *35S:NRT1.1-MYC* line[44] (supplementary Fig. 7). As expected, the *NRT1.1* overexpression line did not show root thermomorphogenesis indicating that overexpressing *NRT1.1* overrides the transcriptional repression of *NRT1.1* by HY5 and thus shows a similarity to *hy5-215*. To assess whether the root thermomorphogenesis phenotypes were linked to N–P level alterations at high ambient temperature, we measured nitrate/nitrite and phosphate levels in the mutants (Fig. 5f, g).

Consistent with the MP-AES results (Fig. 1g, i), Col-0 and *hy5-215* showed different patterns of P level alteration at high ambient temperature, while all the genotypes showed similar patterns of decreased N level at high ambient temperature. Interestingly, *chl1-5* and *hy5-215 chl1-5* double mutant plants showed a similar P pattern with *hy5-215*, which indicates that the *NRT1.1* alteration upon temperature is critical for P level alteration at high ambient temperature. To investigate whether the temperature dependent alteration of phosphate levels is a general part of thermomorphogenesis or tied to the *HY5-NRT1.1* module, we utilized the *pils6-1* mutant and the *35S:PILS6-GFP*, as PILS6 is known regulator of root thermomorphogenesis[45]. Interestingly, both *pils6-1* mutant and *PILS6* overexpression line showed similar phosphate levels to Col-0, indicating that P level regulation is not a general part of thermomorphogenesis but is specifically regulated by the *HY5-NRT1.1* module (Supplementary Fig. 8). Taken together, these data suggest that *HY5-NRT1.1* regulatory mechanism has an important role to regulate root thermomorphogenesis and P level alteration at high ambient temperature.

### The *HY5-NRT1.1* regulatory mechanism alters global gene expression at high ambient temperature

Because the transcript level of *NRT1.1* was down-regulated by HY5 in a root specific manner at high ambient temperature, we performed RNAseq to examine the global gene expression pattern at high ambient temperature using Col-0, *hy5-215*, and *chl1-5* plants. These plants were grown at either 21 °C or 28 °C for additional 5 days after 4 days of germination at 21 °C. Consistent with our qPCR results, *NRT1.1* was among the most downregulated gene in Col-0 roots when comparing 28 °C to 21 °C (Fig. 6a).

Many genes were differentially expressed in these three genotypes in response to temperature: 1512, 1761, and 1295 DEGs in Col-0, *chl1-5*, and *hy5-215*, respectively (Fig. 6b). More than 83% of root Col-0 DEGs were shared between Col-0 and *chl1-5*, indicating that despite the hypersensitive root thermomorphogenesis *chl1-5*, the mutant mounts a very similar response. When conducting a Gene Ontology (GO) analysis of the 249 *NRT1.1* dependent genes (DEGs in Col-0 but not in *chl1-5*), the significant GO-term "response to chemical" was significantly enriched. Also, almost half of the root Col-0 DEGs were *HY5* dependent (755 genes out of 1512 DEGs, >49.9%). This set of genes was enriched for the GO categories related to response to nutrients such as nitrate and sucrose. We reasoned that set of thermomorphogenesis genes would be genes that were not differentially expressed in *hy5-215* mutants (as there is no root growth response to temperature) but still occurs in *chl1-5* (this mutant as well as the *hy5-215/chl1-5* double mutant still shows a root growth response to elevated temperature). This was a set of 593 genes, and it was enriched for nutrients responses such as nitrate and sucrose, similar to the 755 DEGs that were *HY5* dependent.

To conduct a more fine-grained analysis of gene expression patterns, we conducted hierarchical clustering and a subsequent GO enrichment analysis of the resulting clusters (Fig. 6c). Cluster 1 contained genes which increased their expression at high ambient temperature for all genotypes, and these were enriched for circadian rhythm and response to heat. This might indicate that circadian rhythm and heat sensing might be less involved downstream of *HY5/NRT1.1* to affect the root thermomorphogenesis and nutrient alteration at high ambient temperature. Cluster 6 contained genes that

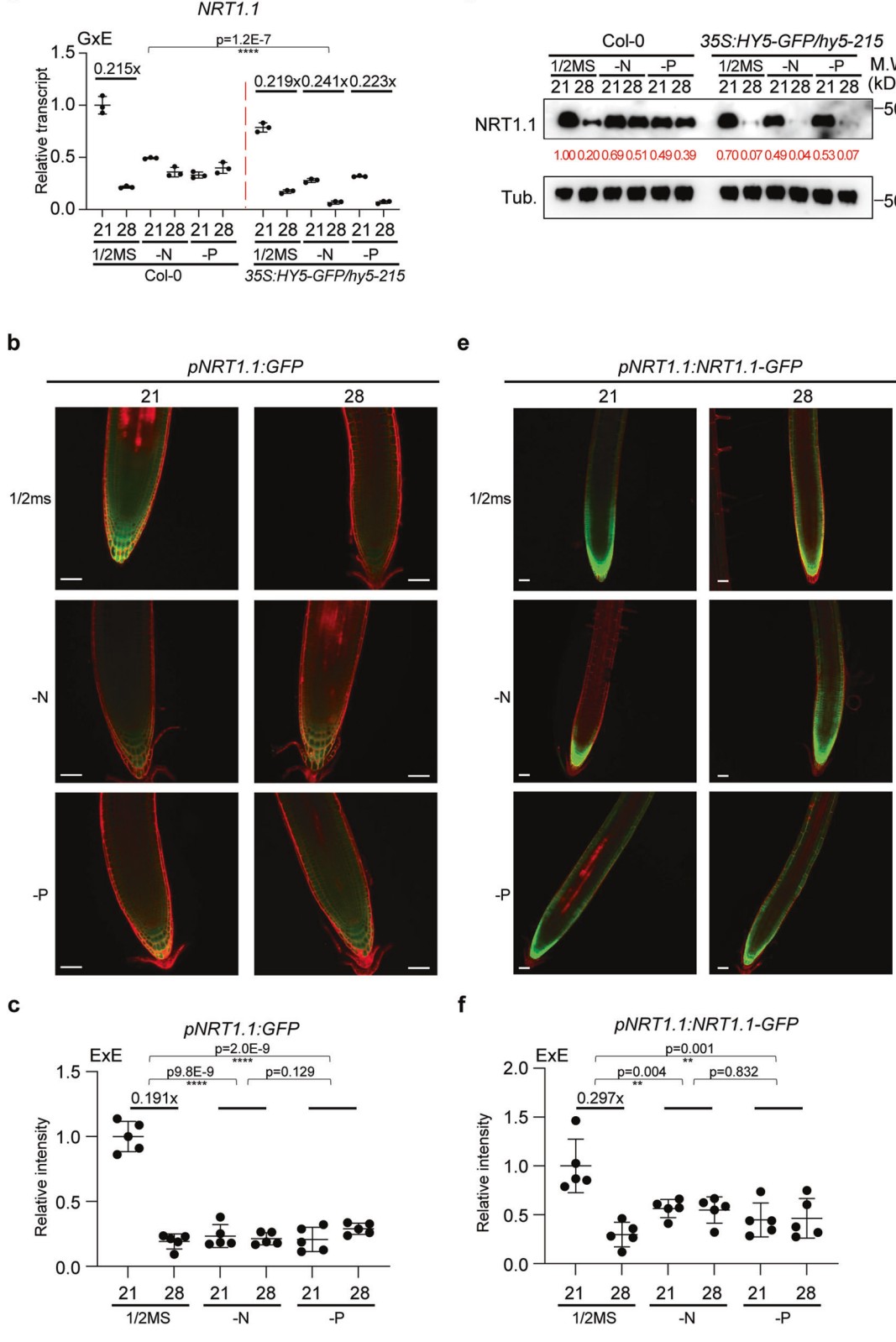

decrease their expression at high ambient temperature in Col-0 and *chl1-5*, but not in *hy5-215*. This gene set was enriched for response to nitrate and inorganic substances. Cluster 7 and 8 contained genes that decrease their expression at high ambient temperature more in *chl1-5* but less in *hy5-215*. This gene set was enriched for suberin biosynthetic process, metabolic processes such as terpenoid or organic acid, and response to water deprivation or abscisic acid. Overall, these data suggest that the *HY5-NRT1.1* regulatory mechanism controls root

thermomorphogenesis and N–P alteration at high ambient temperature through the control of distinct biological processes.

**The *HY5-NRT1.1* regulatory mechanism might be conserved in soybean and rice at higher temperature**

It was previously reported that HY5 and NRT1.1 homologs share similar functions in other species[26,46,47]. Moreover, we showed that root growth responses to higher temperatures are conserved in

**Fig. 4 | NRT1.1 integrates N–P signaling and root thermomorphogenesis.**
**a** Scatter dot plot of qPCR results at normal and high ambient temperature grown Col-0 and *3SS:HY5-GFP/hy5-215* seedlings grown on different media (1/2MS, -N, -P). Only root samples were used for the analysis. The relative transcript level of *NRT1.1* was normalized by the expression levels of *PP2A* and to the expression levels in the shoot. For (**a**), *n* = 3 biologically independent samples were used. **b** Confocal microscopy images of *pNRT1.1:GFP* transgenic line at normal and high ambient temperature grown on different media (1/2MS, -N, -P). **c** Scatter dot plot of the signals quantified from confocal microscopy images of (**b**). Quantification of the signal intensity. **d** Western blot analysis of Col-0 and *3SS:HY5-GFP/hy5-215* seedling roots using native NRT1.1 antibody. Red number indicates the relative signal intensity divided by NRT1.1 signal to Tubulin. Two independent experiments were repeated with similar results. **e** Confocal microscopy images of *pNRT1.1:NRT1.1-GFP* transgenic line at normal and high ambient temperature grown on different media (1/2 ms, -N, -P). Scale bar indicates 50 μm. **f** Scatter dot plot of the signals quantified from confocal microscopy images of (**e**). Media concentrations includes: 1/2MS (N:11400 μM, P: 625 μM), mildly nitrogen deficient (N: 550 μM, P: 625 μM), and mildly phosphorus deficient (N:11400 μM, P: 100 μM). Scatter dot plots indicate mean (horizontal line) and standard deviation (error bars). For (**c** and **f**), *n* = 5 biologically independent samples were used. *p*-Values for the corresponding ExE interactions determined through ANOVA are shown on top of each graph. Asterisks indicate statistically significant difference either 2-way ANOVA; *$p < 0.05$, **$p < 0.01$, ***$p < 0.001$, and ****$p < 0.0001$.

Arabidopsis, soybean and rice (Fig. 1c, e). To explore whether the *HY5-NRT1.1* regulatory mechanism that we had discovered to be responsible for this in Arabidopsis, is conserved in soybean and rice at higher temperatures, and potentially responsible for the root growth response to higher temperature, we analyzed homologs of *HY5* and *NRT1.1* in soy and rice (Fig. 7a, b, Supplementary Fig. 9, 10). While soy has 4 homologs of both genes, *HY5* and *NRT1.1*, rice has 3 and 2 homologs of *HY5* and *NRT1.1*, respectively. To explore the expression pattern of these homologs in response to high temperature, we performed qPCR (Fig. 7c, d). While all of the 4 *HY5* homologs in soybean showed a higher transcript level in the root at higher temperature, in rice, only *OsbZIP48* (*OsHY5L2*) showed increased transcript levels in the root at higher temperatures. This might suggest that while all of 4 *HY5* homologs in soy have a role at higher temperature, only *OsbZIP48* might be involved in the response to higher temperature response in rice. Intriguingly, 2 *NRT1.1* homologs in soy showed decreased transcript levels in the root at higher temperature, and the other 2 *NRT1.1* homologs in soy showed increased transcript levels in the root at higher temperatures. This might suggest that the function of these genes might have diverged or neo-functionalized with regards to their role at higher temperatures. In rice, only one of 2 *NRT1.1* homologs showed a decreased transcript level in the root at higher temperature. Overall, this suggests that the role of *HY5-NRT1.1* homologs in soy is more complex, whereas *OsbZIP48-OsNRT1.1A* might be responsible for higher temperature response in rice. We also performed Western blot analysis using native Arabidopsis NRT1.1 and HY5 antibodies in soy and rice roots (Fig. 7e). Multiple protein bands were detected in soy and rice roots by these antibodies, suggesting that these bands were NRT1.1 and HY5 homolog proteins. Two of the HY5 protein bands in soy and rice, and one of the NRT1.1 protein bands in soy and rice showed a similar pattern to Arabidopsis: NRT1.1 decreasing at higher temperatures and HY5 accumulating at higher temperature. Taken together, our data suggest that *HY5-NRT1.1* regulatory mechanism might be conserved in plants and may contribute to regulate thermomorphogenesis.

## Discussion

Thermomorphogenesis has been extensively studied in Arabidopsis. We have now shown that this response is conserved in other plant species, covering the major groups of angiosperms, monocots and dicots. We have also found a conserved interaction of thermomorphogenesis and nutrient levels. This interaction is multifaceted: thermomorphogenesis affects the nitrogen and phosphate content of plants, and it is itself dependent on sufficient levels of external nitrogen and phosphate. While the dependence of thermomorphogenesis on external nitrogen and phosphate levels is clearly governed by a *HY5-NRT1.1* regulatory mechanism, only the effect of thermomorphogenesis on phosphate contents of plants seems to be dependent on the *HY5-NRT1.1* mechanism. It is therefore not yet clear how and to which extent the thermomorphogenesis related changes of phosphate and nitrogen tissue levels and the modulation of thermomorphogenesis by the external levels of phosphate and nitrogen are related.

The bZIP transcription factor HY5 was originally identified as a positive regulator of light signaling, and only recently emerged as a major factor to regulate temperature dependent signaling in the root. HY5 functions as both, transcription activator and repressor[11,32,38,48], and regulates numerous genes by binding to its target promoter region; its binding to a distinct set of target promoters is increased at high ambient temperature[6] (Fig. 2a–e). Mutation of *HY5* has wide ranging consequences for nutrient levels, not only for the levels of N and P, but also for those of Ca, Carbon, K, Na, Mn, and Zn (Supplementary Fig. 1). It will be interesting to further elucidate how HY5 changes its preferred direct targets at high ambient temperature.

Downstream of *HY5, NRT1.1* acts as a main player in the interaction of temperature responses and nutrients. NRT1.1 was originally discovered as a dual-affinity nitrate transporter and has been shown to integrate nitrate and phosphate signaling in plants[23–26]. Our data show that decreases in both transcript and protein level of NRT1.1 are critical to trigger root responses and a P level alteration at higher temperatures. In a somehow surprising manner, the HY5-NRT1 regulatory module seems to affect plant tissue P-levels in response to higher temperatures but not N-levels. This is even more surprising as we found that HY5 directly transcriptionally regulates NRT1 and other N-homeostasis genes, but we found no clear signature of a notable direct regulation of P-homeostatic genes.

The root response and alterations in N–P levels are conserved in Arabidopsis, soybean, and rice (Figs. 2, 4, 5, 7). Therefore, it might be interesting to further investigate the genetic relationship among the homologs between NRT1.1 and HY5 in soybean and rice. It seems reasonable to assume that since NRT1.1 shares high homology among plants[49], the role of NRT1.1 might be conserved among plant species at higher temperatures. Because Arabidopsis NRT1 family members have similar but diverged functions[49–51], it is possible that different NRT1s or even other NRTs might have distinct roles at higher temperature. For example, *NRT2.1* has been known as a direct target of HY5[27], however, while *NRT1.1* is strongly regulated (bound and altered transcription level) by HY5 at high ambient temperature in the root (Fig. 2a-c, supplementary Fig. 4), *NRT2.1* is not. This seems to be a similar situation as found in the PIFs. All PIFs interact with PHYTOCHROME B (PhyB) and share a redundant role in the red-light signaling pathway. However, PIF1 has a dominant role in seed germination and chlorophyll biosynthesis[9,52], while PIF4 and PIF7 have dominant roles in shoot thermomorphogenesis[10,13], and PIF4 and PIF5 in shade avoidance response[53].

Overall, the observed conserved interactions of nutrient levels and temperature responses are relevant in several ways when it comes to the ongoing climate change. Global warming will affect soil temperatures profoundly, and it has been shown that soil warming affects nutrient availability, especially N and P[54,55]. It will therefore be important to better understand the interactions of N, P and temperature that relate to plant growth. This becomes even more important as fertilizer production and application (in particular for N) causes significant emissions of greenhouse gases[56,57]. While our nutrient analysis is from 9-day media grown plants 4-week soil grown plants (Fig. 1g-i; Supplementary Fig. 1, 2), longer growth and analysis of nutrients in other plant parts such as

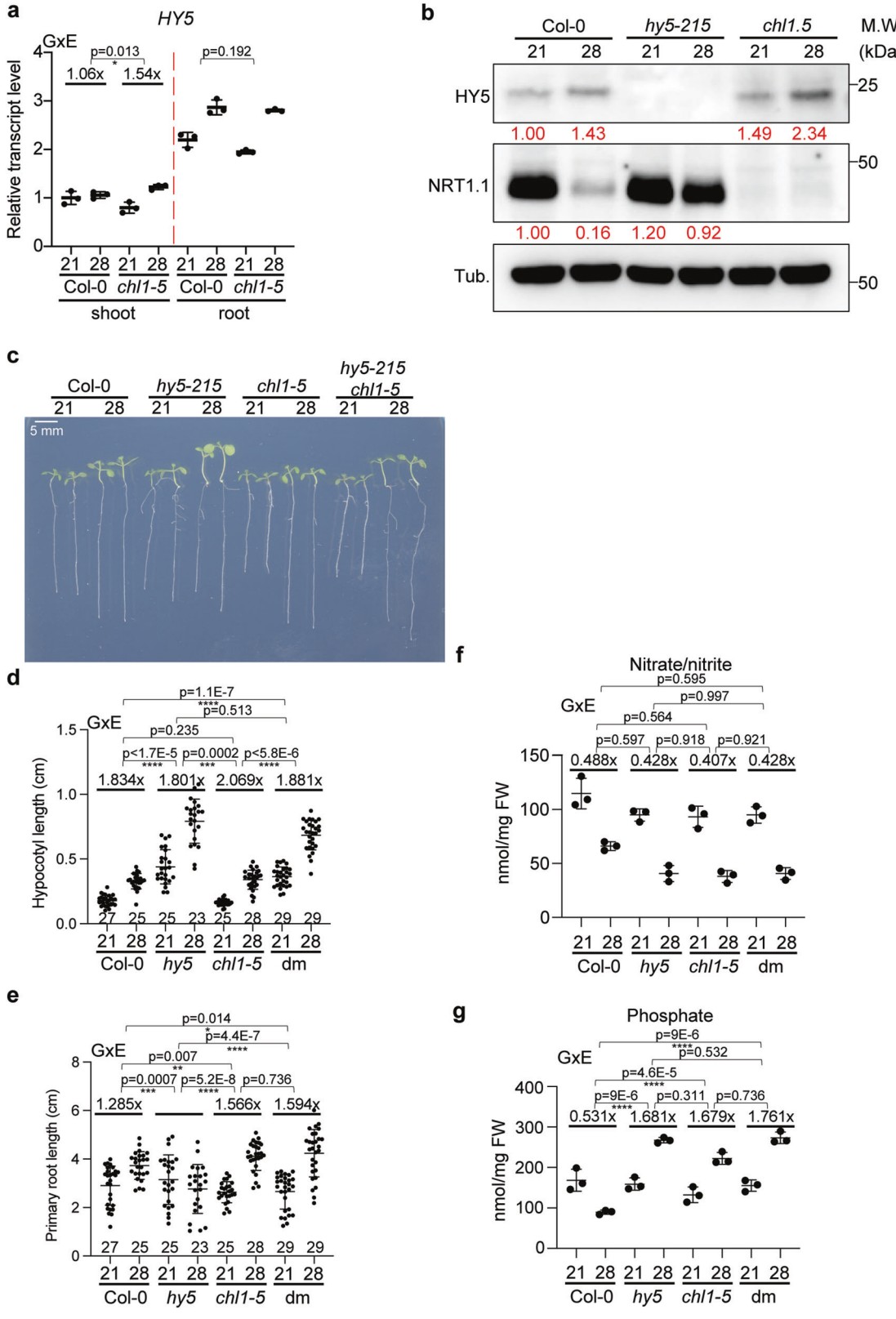

seeds might reveal additional nutrient level changes, some of which might be relevant for the nutritional value of crops. Also, effects on other nutrients at higher temperatures might differ from plant species to plant species, perhaps associated with crop-specific fertilizer formulations[58,59]. Taken together, biotechnology or breeding strategies for taking into considerations future temperature regimes and nutrient levels will be important for overcoming the challenges that global warming will pose for the production of nutritious food and feed at the scale that is needed for a growing global population.

## Methods

### Plant materials, growth conditions, and phenotypic analyses

Arabidopsis plants were grown at LED light with light intensity of 100 μmol and diurnal condition of 16L:8D. Soy and rice plants were

**Fig. 5 | The *HY5-NRT1.1* regulatory module is required for the interaction of root thermomorphogenesis and P level. a** qPCR results of *HY5* transcript level at normal and high ambient temperature using Col-0 and *chl1-5* seedlings with separated samples of shoot and root. Relative transcript level was normalized using *PP2A* as a control and to the expression levels in the shoot. Shoot and root samples were analyzed separately. *n* = 3 biologically independent samples were used. **b** Western blot analyses of NRT1.1 and HY5 using Col-0, *hy5-215*, and *chl1-5* root. Red number indicates the relative signal intensity divided by NRT1.1 or HY5 signal to Tubulin. **c** Phenotypic analyses of Col-0, *hy5-215*, *chl1-5*, *hy5-215 chl1-5* double mutant at high ambient temperature. **d**–**g** Scatter dot plot of phenotypic analyses,

hypocotyl length (**d**), root length (**e**), nitrate and nitrite composition (**f**), and phosphate composition (**g**). For nitrate/nitrite and phosphate composition analyses, seeds of each genotype were grown on soil for 2 weeks at 21 °C and transferred into either 21 °C or 28 °C for additional 2 weeks. Then the leaves were used for the analyses. For (**f** and **g**), *n* = 3 biologically independent samples were used. *p*-Values for the corresponding GxE interactions determined through ANOVA are shown on top of each graph. Asterisks indicate statistically significant difference either 2-way ANOVA; *$p < 0.05$, **$p < 0.01$, ***$p < 0.001$, and ****$p < 0.0001$. Average fold difference of each group is indicated in the top region of the plot. Scatter dot plots indicate mean (horizontal line) and standard deviation (error bars).

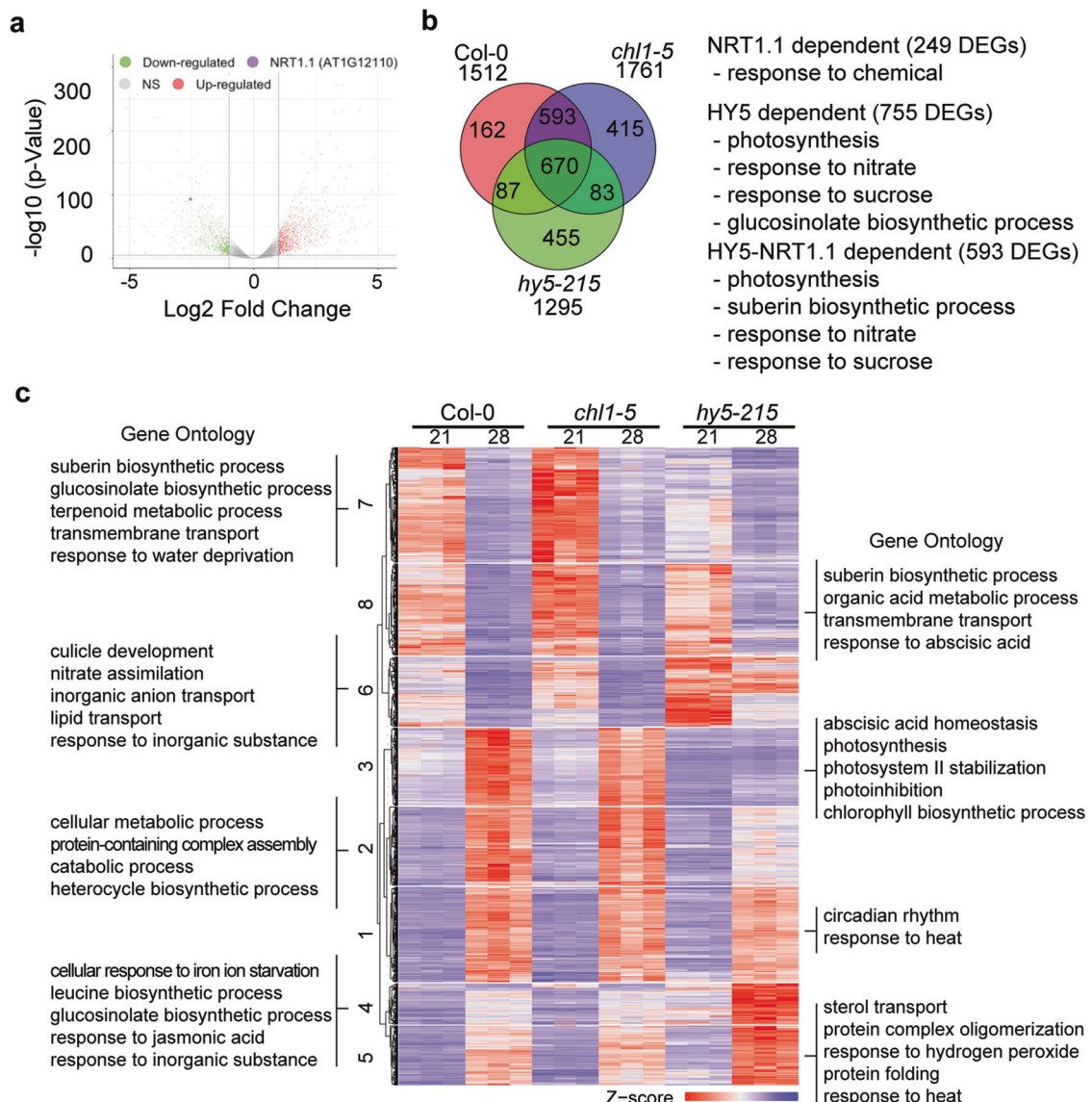

**Fig. 6 | *HY5-NRT1.1* regulatory mechanism controls global gene expression at high ambient temperature. a** Volcano plot of RNAseq using Col-0 root samples. Col-0 seedlings were grown for 4 days in 21 °C and then transferred to either 21 °C or 28 °C for additional 5 days and collected. Threshold of two sided *p*-value is 0.05, and Log2 Fold change threshold is −1 and 1. *NRT1.1* is labeled with purple dot. **b** Venn diagram of root Differentially Expressed Genes (DEGs) in Col-0, *chl1-5*, and

*hy5-215* mutant at high ambient temperature. Gene Ontology (GO) analysis clusters within each group are described next to the Venn diagram. **c** Heatmap analysis shows that Col-0, *chl1-5*, and *hy5-215* mutant have different expression pattern. Representative GO analyses of each cluster are noted. Three biological repeats were performed for RNAseq analysis.

grown at LED light with light intensity of 200 μmol and diurnal condition of 16L:8D. Arabidopsis seedlings were plated in the media and grown vertically at either 21 °C or 28 °C for additional 5 days after 4 days of germination at 21 °C. For the media preparation, MS full media (Caisson lab, Cat. MSP33) Nitrogen-deficient media (Caisson lab,

Cat. MSP19), or Phosphate deficient media (Caisson lab, Cat. MSP21) were used with pH 5.7, 0.8% micropropagation Type 1 phytoagar (Caisson), and supplement of nitrogen or phosphate source according to the previous report[41] (N550: 50μM $NH_4NO_3$, 450μM $KNO_3$, and 8900μM KCl, P125: 100μM $KH_2PO_4$ and 525μM KCl). And then plates

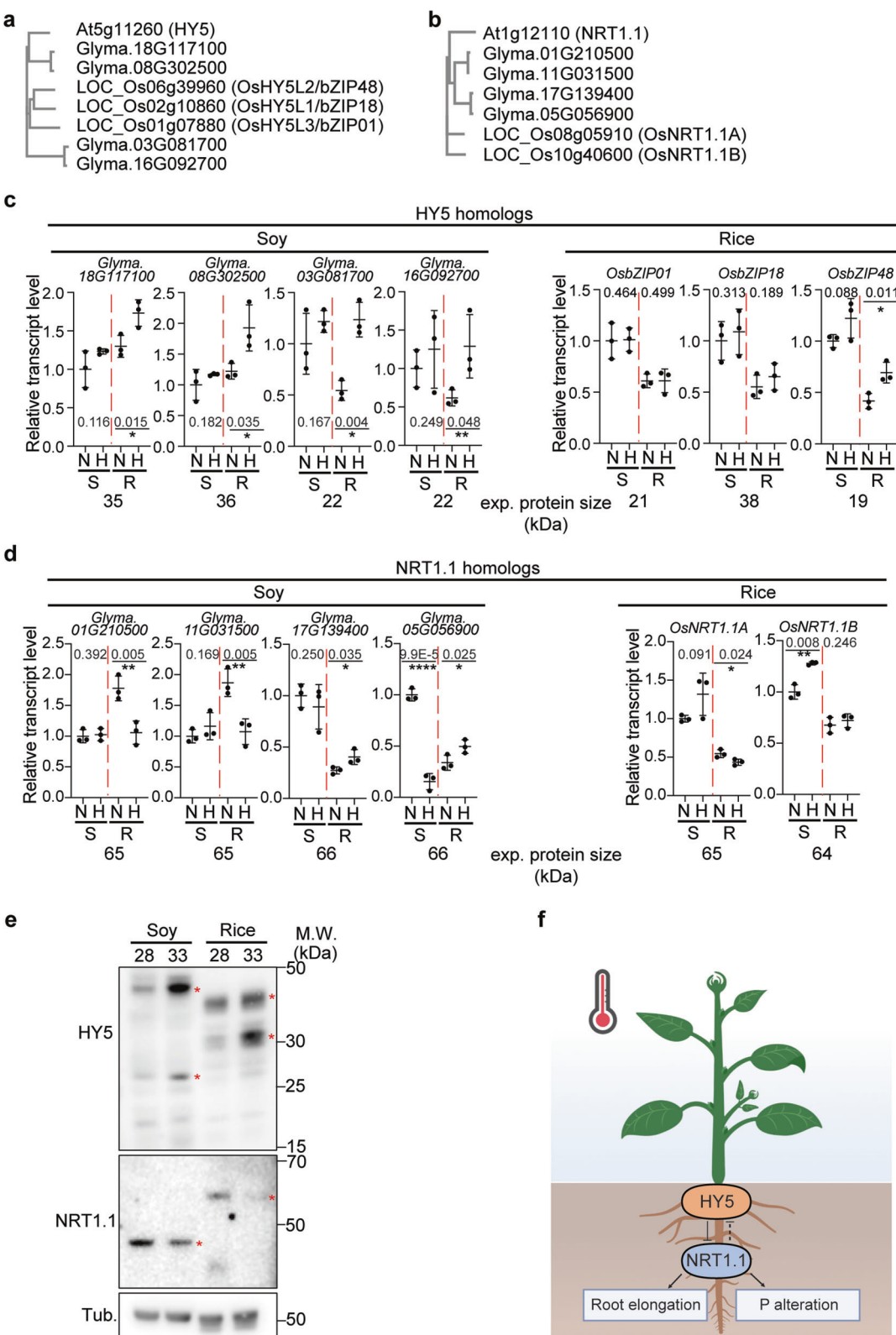

were scanned through the scanner (Epson, Perfection v600) for further analysis using imageJ. The total root length of soy and rice roots was measured using Rhizovision with the parameter: broken roots mode with image threshold 200[60].

### Nutrient analyses

C/N analysis and MP-AES analysis were conducted in this study. For C/N analysis, tissue samples were dried in 70 °C oven for 48 h. Then

Genogrinder (Spex SamplePrep 2010-115) was used for milling the samples. Powder samples were sent to NuMega Resonance Labs (San Diego, CA) for Perkin Elmer PE2400-Series II, CHNS/O analyzer analysis. For MP-AES analysis, Dry samples (~5 mg) were digested with nitric acid 65% (EMD Millipore Cat. 1.00456.2500) and hydrogen peroxide 30% (Sigma-Aldrich Cat. H3410-1L) using an Environmental Express® Hotblock digestion system (Cat. SC196). After diluting the samples with miliQ water, they were quantified by 4210 MP-AES (Agilent).

**Fig. 7 | The *HY5-NRT1.1* regulatory mechanism might be conserved in soybean and rice at higher temperature. a, b** Phylogram of HY5 (**a**) and NRT1.1 (**b**) homologs in Arabidopsis, soy, and rice. **c, d** Relative transcript level of *HY5* (**c**) and *NRT1.1* (**d**) homologs in soy and rice. Relative transcript level is normalized by house-keeping genes such as rice *ubiquitin* and soy *tubulin 4*. N and H stand for Normal temperature (28 °C) and Higher temperature (33 °C), respectively. S and R stand for Shoot and Root, respectively. For (**c** and **d**), $n = 3$ biologically independent samples were used. Asterisks indicate statistically significant difference using one-sided Student's $t$ test; $*p < 0.05$, $**p < 0.01$, $***p < 0.001$, and $****p < 0.0001$. **e** Western blot analysis using native Arabidopsis NRT1.1 and HY5 antibodies.

Soybean and rice root samples were used. Red asterisks are temperature dependent bands which might be potential HY5 and NRT1.1 bands in soybean and rice. Two independent experiments were repeated with similar results. **f** Simplified model showing root specific HY5 accumulation inhibiting NRT1.1 transcription to promote root elongation and the HY5-NRT1.1 regulatory mechanism altering N and P uptake at higher temperatures in plants. Figure 7f, created with BioRender.com, released under a Creative Commons Attribution-NonCommercial-NoDerivs 4.0 International license. Scatter dot plots indicate mean (horizontal line) and standard deviation (error bars).

The following wavelengths were used: 202.548 nm for Zn, 214.915 nm for P, 280.271 nm for Mg, 327.395 nm for Cu, 371.993 for Fe, 403.076 nm for Mn, 589.592 nm for Na, 616.217 nm for Ca, 769.897 nm for K. The final concentration was determined using a standard curve.

## Confocal microscopy

For the confocal microscopy experiments, plants with two genotypes, *pNRT1.1: GFP* and *pNRT1.1:NRT1.1-GFP* transgenic lines were grown at either 21 °C or 28 °C for an additional 5 days after 4 days of germination at 21 °C. Zeiss LSM710 confocal microscope was used for the experiment using the 10x or 20x. Software Zeiss Zen was used for the analysis. For the PI staining, Propidium Iodide (Sigma-Aldrich, Cat. 4170) was used. Samples were gently placed on the solution and stained. For the laser and the filter, a 514 nm laser and a 520/570 nm filter for GFP. The PI signal is excited with either 488 or 514 nm laser and fluorescence emission was filtered by a 600/650 nm filter.

## Protein extraction and Western blot analyses

Total protein extracts were made from 50 seedlings of root sample using 50 µL protein extraction buffer, consisting of 0.35 M Tris-Cl pH 7.5, 10x NuPAGE Sample Reducing Agent (Thermofisher, Cat. NP0009), and 4x NuPAGE LDS Sample Buffer (Thermofisher, Cat. NP0008), and 1× protease inhibitor cocktail. After boiling the samples, samples were centrifuged at $20,200 \times g$ for 5 min and loaded into SDS-PAGE gels (Invitrogen, Cat. NP0323BOX). Separated proteins were transferred using transfer stacks (Invitrogen, Cat. IB23002) and then immunoblotted using anti-HY5 (Abiocode, Cat. R1245-2, dilution 1:3000), anti-NRT1.1 (Agrisera, Cat. AS12 2611, dilution 1:2000), or anti-Tubulin (Invitrogen, Cat. 32-2500, dilution 1:5000) antibodies. For the secondary antibodies, anti-mouse (Biorad, Cat. 170-6516, dilution 1:5000), anti-rabbit (Agrisera, Cat. AS09 602, dilution 1:5000), or were used. SuperSignal West Femto Chemiluminescent substrate (ThermoFisher Scientific, Cat. PI34094) was used for the detection of signals.

## RNA extraction, cDNA synthesis, and qRT-PCR

For RNA sample preparation, plants with three genotypes, Col-0, *hy5-215*, and *chl1-5* mutant, were grown at either 21 °C or 28 °C for additional 5 days after 4 days of germination at 21 °C. Shoot and root separated samples were collected with three independent replicates. Samples were extracted with the RNeasy Plant Mini Kit (Qiagen, Cat. 74904). Thermo Scientific™ Maxima H Minus First Strand cDNA Synthesis Kit (ThermoFisher, Cat. FERK1652) was used for cDNA synthesis. Luna® Universal qPCR Master Mix (NEB, Cat. M3003L) and qPCR machine (Biorad, CFX Opus 384 Real-Time PCR System) were used for qPCR analysis. Primer list is in Supplementary Data 2.

## RNA-seq analyses

For RNAseq sample preparation, same condition was used as qRT-PCR. The RNA quality and quantity were analyzed using a 2100 Bioanalyzer tape station (Agilent Technologies) and Qubit Fluorometer (Invitrogen). The sequencing libraries were generated by the Salk Next Generation Sequencing Core according to Illumina manufacturer's instructions. Sequencing was performed using the Illumina Novaseq6000 platform. For RNAseq analysis, we mapped the short-reads

using Arabidopsis Information resource web site (http://www.arabidopsis.org)[61] combined with the Splice Transcripts Alignments to Reference (STAR) version 2.7.0a method[62]. Differentially Expressed Genes (DEG) analysis was performed using edgeR[63]. Critical values for the analysis are a false discovery rate (FDR < 0.05) and log2FC (> 0 or <0). DEGs were visualized and k-means clustering method was used to classify DEGs (FDR < 0.05 and |log2FC |>1) by comparing the 21 °C and 28 °C treatment for Col-0 in roots via the ComplexHeatmap[64] package in R. The cluster number (k = 8) was determined by sum of squared error and Bayesian information criterion. The volcano plot was created using Enhanced Volcano R package (https://github.com/kevinblighe/EnhancedVolcano). Raw data and processed data for RNA-Seq in Col-0, *hy5-215*, and *chl1.5* can be accessed from the Gene Expression Omnibus database under accession number GSE262197.

## Chromatin immunoprecipitation (ChIP) assays

Chromatin Immunoprecipitation (ChIP) assays were conducted as described previously with minor modifications[6]. For ChIP assay samples, 4-day-old seedlings of *pHY5:HY5-GFP* were transferred to 21 °C or 28 °C for additional 5 days and then harvested. Samples were crosslinked using 1% formaldehyde under 30 min of vacuum and 1 M glycine was added for an additional 5 min for quenching. Samples were gently washed with distilled water for five times and ground thoroughly with mortar and pestle using liquid nitrogen. All the buffers for ChIP were from ChIP assay kit (Millipore, Cat. 17-295). Ground samples were placed into 1.5 mL microtube with nuclei isolation buffer for 15 min and then centrifuged at $20,200 \times g$ for 10 min at 4 °C. 1 mL lysis buffer was used for resuspension of the pellet. Then, sonication of chromatin pellet was performed using digital sonifier (Fisher Scientific, Sonic Dismembrator Model 500). For immunoprecipitation, ChIP grade anti-GFP (Abcam, Cat. ab6556) and dynabeads were used. After washing series of low salt wash buffer, high salt wash buffer, LiCl wash buffer, and TE buffer, we add elution buffer for elution. Samples were incubated overnight at 65 °C for reverse crosslinking after adding NaCl with a final 0.2 M concentration. Finally, PCR purification kit (QIAGEN) was used for DNA purification after proteinase K treatment for 2 h. Samples without IP were used as input DNA. Enrichment (% of input) was calculated from each sample relative to their corresponding input. Primer list is in Supplementary Data 2.

## EMSA and dual luciferase assay

Electrophoretic Mobility Shift Assay (EMSA) was conducted according to manufacturer's protocol (Thermofisher). The Arabidopsis *HY5* coding sequence (CDS) was cloned into pDEST15 and transformed into E. coli strain BL21. After induction with 0.5 mM IPTG, protein was purified using GST purification kit (Thermofisher). G-box containing primer were labeled with biotin in the 3'end through Eton Bioscience.

Dual luciferase assay was conducted following the manufacturer's protocol (Promega, Cat. N1610). *NRT1.1* promoter region and *HY5* CDS region were cloned into pGreenII 0800-LUC and pGreenII 62-SK, respectively. After transformation into *Agrobacterium* strain GV3101, culture medium was resuspended into infiltration buffer (10 mM MgCl$_2$, 10 mM MES and 200 µM acetosyringone) with adjustment of Optical Density (O.D.) 0.2 and incubated for 2 additional h. Subsequently, medium was infiltrated into *Nicotiana Benthamiana* leaves for

transient expression. After 48 hrs, leaves were collected and FLUC and RLUC were measured using the Tecan Safire 2 platereader. The primer list is in Supplementary Data 2.

## Reporting summary

Further information on research design is available in the Nature Portfolio Reporting Summary linked to this article.

## Data availability

RNA sequencing data were deposited into the Gene Expression Omnibus database (accession number GSE262197). Source data are provided with this paper.

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

## Acknowledgements

The authors thank all the Busch lab members for critical discussions. We also thank Dr. Gabriel Krouk for providing *pNRT1:NRT1-GFP* seeds and helpful suggestions. We also thank Drs. Jürgen Kleine-Vehn and Shuichi Yanagisawa for providing *pils6-1* mutant and PILS6-OE, and NRT1.1-OX seeds, respectively. The research was supported by funds from the Salk Harnessing Plants Initiative to W.B. and funds from Michigan State University to H.R.

## Author contributions

S.L. and W.B. conceived the study and designed the experiments. S.L. and J.S. performed root phenotypic analyses. MP-AES analyses performed by G.C. RNAseq analyses performed by S.L. and L.Z. S.L. was responsible for all other experiments. All the authors analyzed the data. W.B. and H.R. supervised work and provided funds and resources. S.L. and W.B. wrote the manuscript with input of all the authors. All the authors discussed the results and commented on the manuscript.

## Competing interests

The authors declare no competing interests.
