## [Peer Review File · Nature Communications]

Nutrient levels control root growth responses to high ambient temperature in plantsReviewer #1 (Remarks to the Author):

The manuscript by Lee et al. deals with root thermomorphogenesis and the genes associated with that response. Authors show that higher ambient temperatures affect root growth with longer roots and this response is seen not only in Arabidopsis, but also in soybean and rice and thus the process is likely to be conserved. Authors show there is thermal response and it is compromised in the hy5 mutants and claim that there is a link between nutrients and HY5/NRT1 is the central component of this response. Through transcriptomics analysis, they claim this process is conserved in Soybean and rice and thus root thermomorphogenesis is a conserved mechanism. By and large it is an interesting study. I do however have some specific concerns that the authors may want to address.

The primary concern for me arises from what appears to me as inappropriate use of statistics throughout this manuscript. For example, one of the key claims that forms the basis of this paper is that the thermomorphogenetic responses are conserved and they are linked to the altered nutrient levels. This claim suffers from the following reasons that I have described below. To show this clearly and to show the involvement of genes such as HY5, a GXE testing is required, which is not done here, and a pairwise difference in one way ANOVA does not really show this. I will illustrate this with one figure, but the same is true for pretty much all figures presented in the manuscript.

Fig 1g-i. The claim made in this figure is that the mutants respond differently compared to wild type to a change in temperature. But the statistics shown is that the hypocotyl length is different in these mutants by Tukey's posthoc test. This is not quite appropriate. What the authors have to show is that there is a differential response to a change in temperature based on genotype, i.e., There is a GxE interaction, which is significant in a two-way ANOVA (NOT ONE WAY ANOVA) with the phenotypes (percentage, C/N ratio, ppm/mg DW) as response and genotype and temperature as factors with interaction terms and show the p-values for the interaction terms. Fig 2, 3, 4, 6 as well as supplementary figures have the same problem. Without a significant GXE (i.e., not parallel reaction norms and a significant interaction) the link is not really shown appropriately. I think a GXE would also allow them to separate the genes where they actually have an effect and those where they really do not have these effects.

A second key concern relates to mixing of shoot and root effects in a subtle but potentially troublesome way. For example, in the ChIP experiments, it is unclear whether the authors did the ChIP experiments using the root tissue or whole seedlings from the description. From the description it looks like it is full seedlings, and it is clear from their phenotypes the effects on shoots is larger than the effects on roots. Therefore, it is unclear to me how a change in expression in root be directly attributed to HY5 binding and regulation? Perhaps it could be, but the text is written in a way that is somewhat misleading. The same comments that I made for Fig 1g-1 apply here as well with Fig 2e.

Minor points.

Fig 1d-f: The effects on Soy and rice in root is not quite that strong and it is unclear how much biological difference do these really make. There are numbers added in some panels and not in others.

- 1) Fig 2f – Is there a temperature nutrient interaction in HY5 expression? Perhaps a qRT would allow to test the GXE interaction significance?
- 2) Fig 6e. Why two bands for HY5 and NRT1.1?

Reviewer #2 (Remarks to the Author):

This manuscript by Lee et al. reports the interesting link between high temperature and nutrient levels through HY5-NRT1.1 module, this might be a conserved feature in monocots and dicots plant species. The authors shown that decreased nitrogen or phosphorus level could mimic hy5 mutant phenotypes under high temperature condition. Low nitrogen or phosphorus supplying highly decreases HY5 protein level in root. Conversely, HY5 specifically down regulates NRT1.1 (master regulator of N and P) expression in root to influence nitrogen and phosphorus contents in plant. In general, the data presented in this manuscript are interesting and contribute to a better understanding of the crosstalk between plant root thermomorphogenesis and nutrients supplying. However, to this reviewer, the molecular mechanism presented in this manuscript is a known regulatory module, several results are not convinced and some analyses and interpretations appear less than rigorous. Below are detailed comments:

1. Figure 1g-i and supplementary Figure1, show the various nutrients content in plant shoot. Consider that this manuscript focuses on root thermomorphogenesis, detect the nutrients content in root is needed. In this assay, why the authors collected "4 week-grown plants " samples rather than" seedlings, grown at either 21°C or 28°C for 5 days after 4 days of germination at 21°C".
2. Line 91-92, the authors proposed that nutrients uptake may change by elevated temperature but they didn't show the results.
3. Figure 2f, the authors have found that SPA kinase activity are involving in HY5-root thermomorphogenesis modulation (Lee et al., 2021). Whether SPA transduces "-N" or "-P" signals to regulate HY5 protein stability. If yes, how? If not, what are the regulators transducing "-N" or "-P" signals and how?
4. Whether "-N" or "-P" treatments affect HY5 gene expression.
5. Figure 2d and e, the primary root length of seedlings (-N(N550,P625) in figure 2d are not the same with the quantification data of (-N(N550,P625)) in figure 2e. The quantification data of "-N(N550,P625), 28" seems even longer than "1/2MS(N11400,P625), 28".
6. Figure 3, it is true that elevated ambient temperature could trigger HY5 accumulation then represses NRT1.1 gene expression level. However, in the case of "-N" or "-P" treatment, HY5 proteins are degraded (Fig 2f), logically it will lead to NRT1.1 inducing regardless of temperature treatment. And the reality is that "-N" or "-P" treatment repress NRT1.1 gene expression, which may indicate that "-N" or "-P" repress NRT1.1 gene expression is independent on elevated temperature and HY5.
7. Line 220-222, the authors claim that "The chl1-5 mutant showed an increased root elongation phenotype at high ambient temperature compared to that of Col-0", however, the results in figure 4c and 4e show that the primary root length of Col-0 and Chl1-5 have no significant difference either in 21°C or 28°C.
8. Figure 1i and Figure 4g, the author's hypothesis is that low phosphorus could mimic hy5 mutant phenotype (Figure 2d and 2e), however, in hy5 and chl1-5 mutants the phosphorus contents are much higher than Col-0 under the treatment of 28°C. How to explain this opposite phenomenon.
9. Figure 4, the labeling letters (a to g) in the figure legend are error.
10. Figure 6, In order to make the convinced conclusion that the thermomorphogenesis is a conserved feature in monocots and dicots plant species, the genetics data of HY5 and NRT1.1 are highly recommended, although HY5 and NRT1.1 in soybean and rice have similar response with Arabidopsis under elevated ambient temperature.

Reviewer #3 (Remarks to the Author):

The interplay between temperature and nutrients is an interesting and important subject. The authors proposed that HY5 and NRT1.1 participate in the link. Nevertheless, several statements and conclusions, replying on correlation, are imprecise or inaccurate, and more (genetic) evidence is needed to support these conclusions.

1. I am very confused by rationale for initiating this study. Figure 1 g and 1i clearly show that HY5 plays a role in regulating the change of phosphate levels BUT NOT NITROGEN LEVELS in response to high temperature. But, then the nitrogen related genes were identified from the CHIP-seq data for further study.

2. Result of Figure 4 c and e clearly show that chl1-5 displays normal thermomorphogenesis response in root growth as "ab" is indistinguishable from "a" in Anova analysis, and therefore CHL1 is not involved in thermomorphogenesis.

3. In Line 155-157 "the lack of temperature dependent root growth in N or P deficient medium is due to the lack of increased HY5 levels". This conclusion relies on the biased view or correlation of the HY5 levels under nutrient deficiency. To have such a strong conclusion, genetic evidence is needed, e.g. overexpressing HY5 or stabilizing HY5 to rescue the loss of thermomorphogenesis under N or P deficiency.

Moreover, how can a slight change (1.3-fold increase) of the HY5 protein in response to high temperature lead to a dramatic change (four-fold decrease) of NRT1.1 expression through direct promoter binding? Does HY5 really directly regulate the expression of NRT1.1? Direct evidence of promoter assay is needed.

4. Despite both dealing with NRT1.1 protein levels, Figure 3d is not consistent with Figure 3f.

5. Statement in Line 176-177 is not correct, as in N deficient media, NRT1.1 transcript level is decreased in response to high temperature in Figure 3a. Therefore, the conclusion of this paragraph in line 183-184, "sufficient amounts of N and P are necessary for the suppression of NRT1.1 transcript level at high ambient temperature," is not accurate. This might be true for P but not N.

6. For Line 197 to 198, the data presented in Figure 3 didn't support this conclusion.

7. For Line 212 to 213, promoter assay is needed to claim that HY5 has a major role to suppress NRT1.1 transcript level.

Minor points:

1. Line 62-63, "Liu 2003 EMBO J" need to be cited for "NRT1.1 encodes for a dual-affinity nitrate transporter", and "Ho 2009 Cell" need to be cited for "NRT1.1 encodes for nitrate sensor".

2. Line 89-90, for "HY5 in nitrogen signaling", "Chen 2016 Cur. Biol." (the important reference for this topic) and "Jonassen EM 2008 Planta" (the earliest reference for this topic) should be cited.

3. Line 94, the reference for "hy5-215 and pifQ" need to be cited.

4. Please carefully check if proper references are cited for the other statements in the introduction, and make sure that the order of the references are cited according to the journal format.

Reviewer #4 (Remarks to the Author):

The report by Lee et al. shows that low external levels of these nutrients abolish root growth responses to high ambient temperature, due to the function of the HY5 transcription factor and its transcriptional regulation of the NRT1.1 gene, proposing the HY5-NRT1.1 regulatory mechanism involved in root thermomorphogenesis.

The interaction between nutrient levels and high ambient temperature response is a new aspect of plant response associated with survival in fluctuating environments. However, I feel the mechanism proposed needs to be supported by more data. For instance, overexpressors of HY5 and NRT1.1 were not analyzed in this study. However, the data with these overexpressors should be shown to support the mechanism proposed. According to the mechanism proposed, constitutively high expression of HY5 should reduce NRT1.1 expression and then root thermomorphogenesis even in (-N) and (-P) media. On the other hand, overexpression of NRT1.1 in Col-0 and hy5 should not cause root thermomorphogenesis at 28°C, like the hy5 mutant. There are also several questions, because HY5 is a transcriptional repressor of NRT1.1. As described in line 191, the observed differences in transcription levels did not reflect protein levels. This reason should be clarified, because the regulation of the NRT1.1 protein level is at the heart of the proposed regulatory mechanism and in this study. Suppose the high NRT1.1 protein levels in (-N) and (-P) media are due to higher translational activity of the NRT1.1 transcript in (-N) and (-P) media or low degradation activity of NRT1.1 protein in (-N) and (-P) media. In that case, it is questionable to what extent HY5-mediated transcriptional regulation of the NRT1.1 gene is essential to the regulation of the NRT1.1 protein level, which was suggested as the key for root thermomorphogenesis. In this connection, comparison of NRT1.1 expression levels in 1/2MS, (-N),

and (-P) media appears to suggest that "sufficient amounts of N and P are necessary for the high level of NRT1.1 transcripts at 21°C and that nutrient deficiency and high ambient temperature independently reduce the NRT1.1 transcript level to the lowest level" but not that "these data suggest that sufficient amounts of N and P are necessary for the suppression of NRT1.1 transcript level at high ambient temperature" (line 183).

A previous report suggested that warm ambient temperature destabilizes PILS6, a repressor of root thermomorphogenesis, thereby increasing nuclear auxin levels and promoting root elongation (Proc. Natl. Acad. Sci. USA, 116: 3893–3898, 2019). Therefore, using mutants and overexpressors of PILS6 as controls would be helpful to show that the HY5-NRT1.1 module is uniquely critical for the interaction between thermomorphogenesis and nutrient levels in roots. Furthermore, HY5 is a well-known shoot-to-root mobile signal, although a study suggested that shoot-root transfer of HY5 is only of secondary importance because of the thermomorphogenic behavior of excised roots (Plant Physiol. 2019, 180:757–766), and another study also reported that HY5 protein mobility is not required in the hypocotyl or for shoot-to-root communication (Plant Communications 1, 100078, 2020). This study also likely indicates that HY5 expressed in roots is critical for root thermomorphogenesis. Therefore, this point should be mentioned by adding new data using grafted seedlings or excised roots.

In summary, I think that this study addresses a new important aspect, but it is still in its infancy.

REVIEWER COMMENTS

We would like to thank all reviewers for their helpful comments. We have conducted several additional experiments, enhanced the data analysis to include Genotype by Environment Interactions via 2-way ANOVA and thoroughly revised the manuscript. We believe that we have addressed all concerns, and that the manuscript has significantly improved. Here we briefly summarize the major changes before we provide a point-by-point response.

1) Considering the suggestions regarding the statistical approach in different Arabidopsis genotypes at different temperatures, we conducted two-way ANOVA for all possible GxE interaction. While the vast of our previous conclusions were supported by this, the GxE interaction P value for the interaction of *hy5* with N composition was not significant anymore. We therefore altered our conclusions with regards to the influence of HY5 on the nitrogen level changes in plant tissues upon high ambient temperature. Nevertheless, as Gene Ontology analysis still showed enrichment of nitrogen related processes in gene sets that were affected by HY5, there is still support for the idea that HY5 alters expression of nitrogen related processes genes.

2) We split Fig 2 into two (Fig 2 and Fig 3) since reviewers suggested to investigate the direct transcriptional regulation of HY5 to NRT1.1 promoter and HY5 stability in -N and -P media. Therefore, Fig 2 highlights direct transcriptional suppression of NRT1.1 by HY5 and Fig 3 shows HY5 stability and movement at high ambient temperature in different media conditions (Fig 3).

3) We tested the direct transcription suppression of NRT1.1 by HY5: We added the data of Electrophoretic Mobility Shift Assay (EMSA), dual luciferase assay, and NRT1.1 transcript level in 3 different forms of HY5 (WT, S36A: phosphor-dead, S36D: phosphor-mimic) overexpressing lines in the *hy5-215* mutant. EMSA data supports the ability of HY5 to bind to the G-box motif of NRT1.1 promoter region *in vitro* (Fig. 2d). Furthermore, dual luciferase assay in *N. benthamiana* results shows that HY5 acts as a transcriptional suppressor of NRT1.1 (fig. 2e).

4) HY5 stability and movement: Using 3 different forms of HY5 (WT, S36A: phosphor-dead, S36D: phosphor-mimic) overexpressing lines in the *hy5-215* mutant background, we now could show that the WT version of HY5 overexpression line did show root thermomorphogenesis in all the conditions including ½ MS, -N, and -P (Fig 3b, c). Interestingly, the HY5 S36D version conferred root thermomorphogenesis in all different media conditions (similar to WT version of HY5) while the HY5 S36A version did not respond to any conditions. This suggests that SPA mediated HY5 phosphorylation is indeed regulating HY5 stability in nutrient deficient condition at high ambient temperature.

5) As it is still controversial whether shoot to root mobile function of HY5 has a major role in root thermomorphogenesis, we added data from excised roots (Fig. 3e) showing that excised Col-0 roots are able to respond to temperature only in ½ MS while excised *hy5* roots cannot. Excised Col-0 root did not respond to temperature at -N and -P conditions similar to non-excised (whole) seedlings. This suggests that shoot to root mobile HY5 signal

has no role or only a minor role, while HY5 in the root has a major role during thermomorphogenesis.

6) We modified the text related with the NRT1.1 transcript levels at temperature and media which is suggested by reviewer 4 “sufficient amounts of N and P are necessary for the high level of NRT1.1 transcripts at 21°C and that nutrient deficiency and high ambient temperature independently reduce the NRT1.1 transcript level to the lowest level” (lines: 251 – 254). Also, we added HY5 overexpression line to see whether HY5 is dependent in nutrient deficient conditions (Fig. 4a, d). The results indicate that while NRT1.1 transcript levels might be independent from HY5 at 21°C, HY5 is able to repress NRT1.1 transcription at high ambient temperature to trigger root thermomorphogenesis (lines: 258 – 259). Furthermore, to further investigate in depth between NRT1.1 transcript level and protein level at ½ MS, -N, and -P media (previously Fig 2, now Fig 3), we added data of treatments with CHX and ES9-17, which are protein synthesis inhibitor and clathrin-mediated endocytosis inhibitor respectively (Supplementary Fig. 6). Overall, Concomitant treatment with CHX and ES9-17 resulted in increased levels of NRT1.1 in all treatments when compared to CHX treatments alone, indicating that the observed change of NRT1.1 protein levels at high ambient temperature and in -N and -P conditions are at least partly due to post-translational regulation and degradation (lines: 283 – 287).

Reviewer #1 (Remarks to the Author):

The manuscript by Lee et al. deals with root thermomorphogenesis and the genes associated with that response. Authors show that higher ambient temperatures affect root growth with longer roots and this response is seen not only in Arabidopsis, but also in soybean and rice and thus the process is likely to be conserved. Authors show there is thermal response and it is compromised in the hy5 mutants and claim that there is a link between nutrients and HY5/NRT1 is the central component of this response. Through transcriptomics analysis, they claim this process is conserved in Soybean and rice and thus root thermomorphogenesis is a conserved mechanism. By and large it is an interesting study. I do however have some specific concerns that the authors may want to address.

The primary concern for me arises from what appears to me as inappropriate use of statistics throughout this manuscript. For example, one of the key claims that forms the basis of this paper is that the thermomorphogenetic responses are conserved and they are linked to the altered nutrient levels. This claim suffers from the following reasons that I have described below. To show this clearly and to show the involvement of genes such as HY5, a GXE testing is required, which is not done here, and a pairwise difference in one way ANOVA does not really show this. I will illustrate this with one figure, but the same is true for pretty much all figures presented in the manuscript.

Fig 1g-i. The claim made in this figure is that the mutants respond differently compared to wild type to a change in temperature. But the statistics shown is that the hypocotyl length is different in these mutants by Tukey's posthoc test. This is not quite appropriate. What the authors have to show is that there is a differential response to a change in temperature based on genotype, i.e., There is a GxE interaction, which is significant in a two-way ANOVA (NOT ONE WAY ANOVA) with the the phenotypes (percentage, C/N ratio, ppm/mg DW) as response and genotype and temperature as factors with interaction terms and show

the p-values for the interaction terms. Fig 2, 3, 4, 6 as well as supplementary figures have the same problem. Without a significant GXE (i.e., not parallel reaction norms and a significant interaction) the link is not really shown appropriately. I think a GXE would also allow them to separate the genes where they actually have an effect and those where they really do not have these effects.

Author response: Thanks for the suggestion. We have now added GxE interaction using a two-way ANOVA throughout the figures.

A second key concern relates to mixing of shoot and root effects in a subtle but potentially troublesome way. For example, in the ChIP experiments, it is unclear whether the authors did the ChIP experiments using the root tissue or whole seedlings from the description. From the description it looks like it is full seedlings, and it is clear from their phenotypes the effects on shoots is larger than the effects on roots. Therefore, it is unclear to me how a change in expression in root be directly attributed to HY5 binding and regulation? Perhaps it could be, but the text is written in a way that is somewhat misleading. The same comments that I made for Fig 1g-1 apply here as well with Fig 2e.

Author response: We thank the reviewer for raising this issue. The qPCR results show that HY5 is required for repressing NRT1.1 transcription at high ambient temperature in roots (Fig. 2c) and that it is sufficient to repress NRT1.1 in the root when overexpressed (Fig. 4a). However, due to technical limitations, the ChIP-qPCR experiments were performed using whole seedling in this study. New data that supports direct binding including EMSA and dual luciferase assay (Fig. 2d, e) can't also be done in the root. Formally, it can't be fully excluded that HY5 binding can be detected in whole seedlings, *in vitro* (EMSA) and in the tobacco system (dual luciferase) but for an unknown reason (even though HY5 and NRT1.1 are expressed in the root) might not occur in the root. We therefore have clarified this in the text. We therefore added in line 166ff: "Taken together, our data strongly suggest that HY5 directly binds to the promoter region of *NRT1.1* and represses its transcription at high ambient temperature. However, as we performed ChIP-qPCR in whole seedlings, EMSA *in vitro*, and the dual luciferase assay in *Nicotiana benthamiana*, it can't be fully excluded that the direct binding of HY5 to the NRT1.1 promoter, for some unknown reason (even though HY5 and NRT1.1 are expressed in the root), might not occur in the root."

Minor points.

Fig 1d-f: The effects on Soy and rice in root is not quite that strong and it is unclear how much biological difference do these really make. There are numbers added in some panels and not in others.

Author response: Throughout this study, the changes of Arabidopsis root lengths at high ambient temperature are around 30%. In soybean and rice, we can observe this response at a similar level or higher degree. (>70% and 30% increase respectively). To make this clear, we have included in Fig 1b, d, and f in the plots: 1) Average fold change of each comparison. 2) p values from one-sided Student's T-test, and 3) number of plants analyzed in this graph.

1) Fig 2f – Is there a temperature nutrient interaction in HY5 expression? Perhaps a qRT would allow to test the GXE interaction significance?

Author response: We have now added qPCR results of HY5 gene expression in -N and -P conditions (Fig. 3a). Using 2-way ANOVA, we found there is a significant interaction between temperature and nitrogen or phosphorus on HY5 expression levels.

2) Fig 6e. Why two bands for HY5 and NRT1.1?

Author response: Previous Fig 6e which is now Fig 7e is a Western blot analysis of Soybean and Rice root samples using Arabidopsis native HY5 and NRT1.1 antibody. According to our homolog analysis, we found several homologous genes in soybean and rice respectively. This might be the reason for the multiple bands in the western blot. We marked the potential bands of temperature dependent protein levels with the red asterisks. We rephrased the text to make this clearer (lines: 405 – 407).

Reviewer #2 (Remarks to the Author):

This manuscript by Lee et al. reports the interesting link between high temperature and nutrient levels through HY5-NRT1.1 module, this might be a conserved feature in monocots and dicots plant species. The authors shown that decreased nitrogen or phosphorus level could mimic hy5 mutant phenotypes under high temperature condition. Low nitrogen or phosphorus supplying highly decreases HY5 protein level in root. Conversely, HY5 specifically down regulates NRT1.1 (master regulator of N and P) expression in root to influence nitrogen and phosphorus contents in plant. In general, the data presented in this manuscript are interesting and contribute to a better understanding of the crosstalk between plant root thermomorphogenesis and nutrients supplying. However, to this reviewer, the molecular mechanism presented in this manuscript is a known regulatory module, several results are not convinced and some analyses and interpretations appear less than rigorous. Below are detailed comments:

1. Figure 1g-i and supplementary Figure1, show the various nutrients content in plant shoot. Consider that this manuscript focuses on root thermomorphogenesis, detect the nutrients content in root is needed.

In this assay, why the authors collected “4 week-grown plants ” samples rather than” seedlings, grown at either 21°C or 28°C for 5 days after 4 days of germination at 21°C”.

Author response: We thank the reviewer for this suggestion. We added nutrient content data using roots from the seedlings, grown at either 21°C or 28°C for 5 days after 4 days of germination at 21°C (Supplementary Fig 2). Interestingly, roots sampled at the seedling stage did not reflect the results obtained from 4 week-old shoots of “soil” grown plants. This might indicate that nutrient accumulate over time (lines: 109 - 118).

2. Line 91-92, the authors proposed that nutrients uptake may change by elevated temperature but they didn't show the results.

Author response: We apologize for the confusion. In this study, we were only able to focus on nutrient composition. Therefore, we removed the word “uptake” here.

3. Figure 2f, the authors have found that SPA kinase activity are involving in HY5-root thermomorphogenesis modulation (Lee et al., 2021). Whether SPA transduces “-N” or “-P” signals to regulate HY5 protein stability. If yes, how ? If not, what are the regulators transducing “-N” or “-P” signals and how?

Author response: We thank the reviewer for this great suggestion. As it had been shown that SPA mediated HY5 phosphorylation is crucial for root thermomorphogenesis, we have included overexpression lines of different HY5 protein versions that relate to SPA function: WT, S36A: phospho-dead, S36D: phospho-mimic overexpressing lines in *hy5-215* mutant in different media and temperature conditions (Fig 3b, c). The WT version of the *HY5* overexpression line showed root thermomorphogenesis in all the conditions including ½ MS, -N, and -P. Interestingly, the *HY5* S36D version was able to show root thermomorphogenesis in all the different media conditions similar to WT version of *HY5* while *HY5* S36A version did not respond to any conditions. This suggests that SPA mediated *HY5* phosphorylation is indeed regulating *HY5* stability in nutrient deficient condition at high ambient temperature as well (lines: 200 – 216).

4. Whether “-N” or “-P” treatments affect *HY5* gene expression.

Author response: This is another great suggestion. We have now added qPCR results of *HY5* gene expression in -N and -P conditions (Fig. 3a). Similar to the *HY5* protein level, *HY5* transcript levels were decreased at 21 in -N and -P media, and did not change at high ambient temperature. This suggests that sufficient N and P are necessary to trigger *HY5* transcription for root thermomorphogenesis (lines: 192 – 194).

5. Figure 2d and e, the primary root length of seedlings (-N(N550,P625) in figure 2d are not the same with the quantification data of (-N(N550,P625)) in figure 2e. The quantification data of “-N(N550,P625), 28” seems even longer than “1/2MS(N11400,P625), 28”.

Author response: We did the same experiment again, also adding 3 different forms of *HY5* (WT, S36A: phospho-dead, S36D: phospho-mimic) overexpressing lines in *hy5-215* mutant and presented the new analysis in the figure, which is now fig. 3c, d.

6. Figure 3, it is true that elevated ambient temperature could trigger *HY5* accumulation then represses *NRT1.1* gene expression level. However, in the case of “-N” or “-P” treatment, *HY5* proteins are degraded (Fig 2f), logically it will lead to *NRT1.1* inducing regardless of temperature treatment. And the reality is that “-N” or “-P” treatment repress *NRT1.1* gene expression, which may indicate that “-N” or “-P” repress *NRT1.1* gene expression is independent on elevated temperature and *HY5*.

Author response: We agree with the reviewer’s point. Therefore, we added the qPCR results of *HY5* overexpression line (Fig. 4a). The results indicate that while *NRT1.1* transcript levels might be independent from *HY5* at 21°C, *HY5* is able to repress *NRT1.1* transcription at high ambient temperature to trigger root thermomorphogenesis (lines: 258 – 259). Also, to make the explanation clearer, we modified the text as also suggested by

reviewer 4 “sufficient amounts of N and P are necessary for the high level of NRT1.1 transcripts at 21°C and that nutrient deficiency and high ambient temperature independently reduce the NRT1.1 transcript level to the lowest level” (lines: 251 – 254).

7. Line 220-222, the authors claim that “The chl1-5 mutant showed an increased root elongation phenotype at high ambient temperature compared to that of Col-0”, however, the results in figure 4c and 4e show that the primary root length of Col-0 and Chl1-5 have no significant difference either in 21°C or 28°C.

Author response: We analyzed the data using 2-way ANOVA for the GxE interaction and found that chl1-5 and Col-0 are statistically different with a P value of 0.007. It is now changed into fig 5c.

8. Figure 1i and Figure 4g, the author’s hypothesis is that low phosphorus could mimic hy5 mutant phenotype (Figure 2d and 2e), however, in hy5 and chl1-5 mutants the phosphorus contents are much higher than Col-0 under the treatment of 28°C. How to explain this opposite phenomenon.

Author response: There might be a number of possible explanations. For example, the underlying mechanism might be different for the response to low external P levels and the tissue P levels. We were not able to conduct experiments to clarify that would fall into the scope of this (already expansive) manuscript. We therefore added to the discussions in line 418ff: “However, it is not yet clear how and to which extent the thermomorphogenesis related changes of phosphate and nitrogen tissue levels and the modulation of thermomorphogenesis by the external levels of phosphate and nitrogen are related.”

9. Figure 4, the labeling letters (a to g) in the figure legend are error.

Author response: We corrected the error in the figure legend which is now Fig. 5.

10. Figure 6, In order to make the convinced conclusion that the thermomorphogenesis is a conserved feature in monocots and dicots plant species, the genetics data of HY5 and NRT1.1 are highly recommended, although HY5 and NRT1.1 in soybean and rice have similar response with Arabidopsis under elevated ambient temperature.

Author response: We thank the reviewer for this suggestion. However, due to more than 1 year that it would take to gene edit these species, we decided that this is out of the scope of this work and is better addressed in future work. We modified the text (lines: 437 – 439).

Reviewer #3 (Remarks to the Author):

The interplay between temperature and nutrients is an interesting and important subject. The authors proposed that HY5 and NRT1.1 participate in the link. Nevertheless, several statements and conclusions, relying on correlation, are imprecise or inaccurate, and more (genetic) evidence is needed to support these conclusions.

1. I am very confused by rationale for initiating this study. Figure 1 g and 1i clearly show that HY5 plays a role in regulating the change of phosphate levels BUT NOT NITROGEN

LEVELS in response to high temperature. But, then the nitrogen related genes were identified from the CHIP-seq data for further study.

Author response: It is true that we found that *HY5* is required for temperature dependent P-level changes and not N-level changes in Arabidopsis. But we had also found that *HY5* affects genes involved in nitrogen related processes at high ambient temperature at least at the transcriptional level. Therefore, when we analyzed ChIP enriched promoter regions possessing genes using Gene Ontology finding many genes related to nitrogen (Fig.2a, b and supplementary fig. 4) but no well-known phosphorus related genes other than *NRT1.1*, which is involved in the integration of N-P signaling. To support this interesting finding, we added supplementary fig. 3 to show that phosphorus related genes such as *PHO1*, *PHT1;8*, *PHT1;9*, and *IPS1* do not have *HY5* enriched promoter region indicating that it is unlikely to be a target of *HY5* (lines: 136 – 138). Therefore, the data point towards the importance of *NRT1.1*, which has been shown to play a role in N-P integration and for which we show that it is critical for P level alteration at high ambient temperature (Fig. 5g).

2. Result of Figure 4 c and e clearly show that *chl1-5* displays normal thermomorphogenesis response in root growth as “ab” is indistinguishable from “a” in Anova analysis, and therefore *CHL1* is not involved in thermomorphogenesis.

Author response: We analyzed the data using 2-way ANOVA for the GxE interaction and found that *chl1-5* and *Col-0* are statistically different with the P value 0.007. It is now changed into fig 5c.

3. In Line 155-157 “the lack of temperature dependent root growth in N or P deficient medium is due to the lack of increased *HY5* levels”. This conclusion relies on the biased view or correlation of the *HY5* levels under nutrient deficiency. To have such a strong conclusion, genetic evidence is needed, e.g. overexpressing *HY5* or stabilizing *HY5* to rescue the loss of thermomorphogenesis under N or P deficiency.

Author response: This is a great suggestion. We have now added the phenotypes of 3 different forms of *HY5* (WT, S36A: phospho-dead, S36D: phospho-mimic) overexpressing lines in *hy5-215* mutant at different media and temperature conditions (Fig 3b, c). WT version of *HY5* overexpression line was able to show root thermomorphogenesis in all the conditions including ½ MS, -N, and -P. Interestingly, the *HY5* S36D version showed root thermomorphogenesis in all the different media conditions similar to the WT version of *HY5*, while *HY5* S36A version did not respond to any conditions. This suggests that SPA mediated *HY5* phosphorylation is indeed regulating *HY5* stability in nutrient deficient conditions at high ambient temperature as well (lines: 200 – 216).

Moreover, how can a slight change (1.3-fold increase) of the *HY5* protein in response to high temperature lead to a dramatic change (four-fold decrease) of *NRT1.1* expression through direct promoter binding? Does *HY5* really directly regulate the expression of *NRT1.1*? Direct evidence of promoter assay is needed.

Author response: This is an excellent suggestion. We have now added data of EMSA and dual luciferase assay (Fig 2d, e). EMSA results show that *HY5* directly binds to the G-box motif of *NRT1.1* promoter. Also, dual luciferase assay results show that *HY5* acts as a transcriptional suppressor of *NRT1.1* (lines: 159 – 172).

4. Despite both dealing with NRT1.1 protein levels, Figure 3d is not consistent with Figure 3f.

Author response: NRT1.1 protein level detected in Fig 4d (previously Fig 3d) were from native NRT1.1 antibody using whole root of Col-0 seedling. Relative intensity of pNRT1.1:NRT1.1-GFP (Fig 4f, previously Fig 3f) was measured at the root apex. There might be other minor NRT1.1 expression in the root besides root apex or tip. We modified the text to specify the root apex observation in the pNRT1.1:NRT1.1-GFP experiment. Overall, although there are some differences between those two figures however still suggests that the NRT1.1 protein level is not responding to temperature at -N and -P media (lines: 265 – 267, 292 – 293).

5. Statement in Line 176-177 is not correct, as in N deficient media, NRT1.1 transcript level is decreased in response to high temperature in Figure 3a. Therefore, the conclusion of this paragraph in line 183-184, “sufficient amounts of N and P are necessary for the suppression of NRT1.1 transcript level at high ambient temperature,” is not accurate. This might be true for P but not N.

Author response: A major point of Fig 3 (which is now Fig 4) is that both NRT1.1 transcript and protein levels are already reduced in N and P deficient media at control temperature conditions, and not changing much at high ambient temperature. To make the explanation clearer, we modified the text as also suggested by reviewer 4 “sufficient amounts of N and P are necessary for the high level of NRT1.1 transcripts at 21°C and that nutrient deficiency and high ambient temperature independently reduce the NRT1.1 transcript level to the lowest level” (lines: 251 – 254). We have added the data of the HY5 overexpression line showed decreased transcript and protein levels of NRT1.1 in all tested nutrient conditions (½ MS, -N, and -P media) at high ambient temperature (Fig. 4a, d), which is in line with the thermomorphogenesis phenotypes in these plants (Fig. 3b, c). Overall, these results indicate that while NRT1.1 transcript levels might be independent from HY5 at 21°C, HY5 is able to repress NRT1.1 transcription at high ambient temperature to trigger root thermomorphogenesis (lines: 256 – 259).

6. For Line 197 to 198, the data presented in Figure 3 didn't support this conclusion.

Author response: This is the follow-up suggestion from previous suggestion. Overall, key observation is that decreases of NRT1.1 transcript and protein levels are pronounced in ½ MS condition at high ambient temperature which goes along with root thermomorphogenesis (lines: 271 – 277). And thus, we concluded that “Overall, these data suggest that sufficient amounts of N and P are necessary for the suppression of NRT1.1 transcript level at high ambient temperature (lines: 265 – 267).” and also “Overall, the data suggested that changes in NRT1.1 transcript and protein level might be necessary for root thermomorphogenesis (lines: 292 – 293).”

7. For Line 212 to 213, promoter assay is needed to claim that HY5 has a major role to suppress NRT1.1 transcript level.

Author response: Based on the new data: EMSA and dual luciferase, we can show that HY5 directly binds and acts as a transcriptional suppressor of NRT1.1 (Fig 2d, e) (lines: 159 – 172).

Minor points:

1. Line 62-63, "Liu 2003 EMBO J" need to be cited for "NRT1.1 encodes for a dual-affinity nitrate transporter", and "Ho 2009 Cell" need to be cited for "NRT1.1 encodes for nitrate sensor".

Author response: This has been corrected with proper citations.

2. Line 89-90, for "HY5 in nitrogen signaling", "Chen 2016 Cur. Biol." (the important reference for this topic) and "Jonassen EM 2008 Planta" (the earliest reference for this topic) should be cited.

Author response: This has been corrected with proper citations.

3. Line 94, the reference for "hy5-215 and pifQ" need to be cited.

Author response: This has been corrected with proper citations.

4. Please carefully check if proper references are cited for the other statements in the introduction, and make sure that the order of the references are cited according to the journal format.

Author response: Thank you so much for the suggestion. We have checked this now and made the necessary changes.

Reviewer #4 (Remarks to the Author):

The report by Lee et al. shows that low external levels of these nutrients abolish root growth responses to high ambient temperature, due to the function of the HY5 transcription factor and its transcriptional regulation of the NRT1.1 gene, proposing the HY5-NRT1.1 regulatory mechanism involved in root thermomorphogenesis.

The interaction between nutrient levels and high ambient temperature response is a new aspect of plant response associated with survival in fluctuating environments. However, I feel the mechanism proposed needs to be supported by more data. For instance, overexpressors of HY5 and NRT1.1 were not analyzed in this study. However, the data with these overexpressors should be shown to support the mechanism proposed. According to the mechanism proposed, constitutively high expression of HY5 should reduce NRT1.1 expression and then root thermomorphogenesis even in (-N) and (-P) media. On the other hand, overexpression of NRT1.1 in Col-0 and hy5 should not cause root thermomorphogenesis at 28°C, like the hy5 mutant.

Author response: This is an excellent point, which we addressed. Considering the overexpressing or stabilizing HY5, we added the phenotypes of 3 different forms of HY5 (WT, S36A: phosphor-dead, S36D: phosphor-mimic) overexpressing lines in hy5-215 mutant at different media and temperature conditions (Fig 3b, c). The WT version of the HY5 overexpression line showed root thermomorphogenesis in all the conditions including

½ MS, -N, and -P. The HY5 S36D version showed root thermomorphogenesis in all different media conditions while HY5 S36A version did not lead to thermomorphogenesis in any condition. This suggests that SPA mediated HY5 phosphorylation is regulating HY5 stability in nutrient deficient condition at high ambient temperature (lines: 200 – 216). We also added the data of NRT1.1 transcript and protein level in 35S:HY5-GFP/hy5-215 root (Fig. 4a, d), which was similar to the Col-0 in ½ MS media at high ambient temperature. Consistent with the phenotype, NRT1.1 transcript and protein levels were reduced in -N and -P media at high ambient temperature in 35S:HY5-GFP/hy5-215, indicating that expression of HY5 regulates NRT1.1 mediated root thermomorphogenesis in N and P deficiency (lines: 242 – 259).

Also, we added the phenotype of NRT1.1 OX (Supplementary Fig. 7), which did not show root thermomorphogenesis. This data supports the model that highly expressed HY5 leads to suppress NRT1.1 transcription at high ambient temperature (lines: 322 – 326).

There are also several questions, because HY5 is a transcriptional repressor of NRT1.1. As described in line 191, the observed differences in transcription levels did not reflect protein levels. This reason should be clarified, because the regulation of the NRT1.1 protein level is at the heart of the proposed regulatory mechanism and in this study. Suppose the high NRT1.1 protein levels in (-N) and (-P) media are due to higher translational activity of the NRT1.1 transcript in (-N) and (-P) media or low degradation activity of NRT1.1 protein in (-N) and (-P) media. In that case, it is questionable to what extent HY5-mediated transcriptional regulation of the NRT1.1 gene is essential to the regulation of the NRT1.1 protein level, which was suggested as the key for root thermomorphogenesis. In this connection, comparison of NRT1.1 expression levels in 1/2MS, (-N), and (-P) media appears to suggest that “sufficient amounts of N and P are necessary for the high level of NRT1.1 transcripts at 21°C and that nutrient deficiency and high ambient temperature independently reduce the NRT1.1 transcript level to the lowest level” but not that “these data suggest that sufficient amounts of N and P are necessary for the suppression of NRT1.1 transcript level at high ambient temperature” (line 183).

Author response: This is an excellent suggestion. We modified the text according to your suggestion (lines: 251 – 254). We have added the data of the HY5 overexpression line showed decreased transcript and protein levels of NRT1.1 in all tested nutrient conditions (½ MS, -N, and -P media) at high ambient temperature (Fig. 4a, d), which is in line with the thermomorphogenesis phenotypes in these plants (Fig. 3b, c). Overall, these results indicate that while NRT1.1 transcript levels might be independent from HY5 at 21°C, HY5 is able to repress NRT1.1 transcription at high ambient temperature to trigger root thermomorphogenesis (lines: 256 – 259).

We also have now added the data of NRT1.1 protein level under different media and temperature conditions with cycloheximide (protein synthesis inhibitor) and ES9-17 (clathrin mediated endocytosis inhibitor) (Supplementary Fig. 6). Concomitant treatment with CHX and ES9-17 resulted in increased levels of NRT1.1 in all treatments when compared to CHX treatments alone, indicating that the observed change of NRT1.1 protein levels at high ambient temperature and in -N and -P conditions are at least partly due to post-translational regulation and degradation (lines: 283 – 287).

A previous report suggested that warm ambient temperature destabilizes PILS6, a repressor of root thermomorphogenesis, thereby increasing nuclear auxin levels and promoting root elongation (Proc. Natl. Acad. Sci. USA, 116: 3893–3898, 2019). Therefore, using mutants and overexpressors of PILS6 as controls would be helpful to show that the HY5-NRT1.1 module is uniquely critical for the interaction between thermomorphogenesis and nutrient levels in roots.

Author response: This is an excellent suggestion. We have now added the data of nutrient levels using *pils6-1* and PILS6OE (Supplementary Fig. 8). There was no difference between mutant and overexpression line in Nitrate/nitrite and Phosphorus composition which suggests that PILS6 is not involved in nutrient composition at high ambient temperature (lines: 333 – 340).

Furthermore, HY5 is a well-known shoot-to-root mobile signal, although a study suggested that shoot-root transfer of HY5 is only of secondary importance because of the thermomorphogenic behavior of excised roots (Plant Physiol. 2019, 180:757–766), and another study also reported that HY5 protein mobility is not required in the hypocotyl or for shoot-to-root communication (Plant Communications 1, 100078, 2020). This study also likely indicates that HY5 expressed in roots is critical for root thermomorphogenesis. Therefore, this point should be mentioned by adding new data using grafted seedlings or excised roots.

Author response: This is an excellent suggestion. We have now added excised root data (Fig. 3e) and consistent with the previous publications, excised Col-0 roots were able to respond to higher temperature only in ½ MS media, not in -N or -P media conditions. Furthermore, excised *hy5* roots were not responding to higher temperature indicating that shoot to root mobile signal of HY5 has a minor role, if any, while HY5 in the root has a major role at thermomorphogenesis (lines: 227 – 235).

Reviewer #1 (Remarks to the Author):

My major criticisms in the previous version of the manuscript was related to the statistical analysis and in the present version of the manuscript, authors have included GXE analysis, which has indicated that the effects on N by temperature does not differ between the genotypes. This also raises the rationale of the focus on NRT1.1 and subsequent data analysis and whether the focus should be on ambient temperature N/P interactions or general effect, but that is something I will leave it to the authors. They have changed the wordings a bit, which I think is still a bit too strong and some areas. Authors are still arguing mostly their points, and present the story they like, perhaps more than what their data shows and I think they could make some attempts to make it a bit more clearer and data-driven. Nevertheless overall, it appears that the authors have addressed most of my comments and I do not have much further to add to the review.

Reviewer #2 (Remarks to the Author):

The revised manuscript has greatly improved, but it still needs further improvement on the following points:

1. Line109-114, Thank you for measuring the nutrient contents in the 9-day old seedling roots. But the results show that the high ambient temperature related phenotypes may not dependent on the nutrient contents changing (eg. N, P levels) and HY5 mutation in 9-day old seedling roots. It is different from the responses of 4 week old plant shoots, which the phenotypes are related to the nutrient contents changing (eg. P level) and HY5 mutation. So it may indicate that there have two different regulatory mechanisms in 9 day old seedlings and 4 week old plants, or it exist two different regulatory mechanisms in the shoot and root under high ambient temperature treatments. Consider the nutrient contents data is a key results to link the high ambient temperature to HY5 and plant phenotypes, the results of nutrient contents in 9-day old seedling shoots and 4 week old plant roots are also needed. The difference of nutrient contents in 9-day old seedling and 4 week old plant could not easily explained by "nutrients accumulate over time", more discussion is needed.
2. Related to figure 3b and c, whether NRT1;1 gene expression level also altered in 35S:HY5-GFP, 35S:HY5 S36A-GFP, and 35S:HY5 S36D-GFP lines under 21 /28 °C treatments.
3. How HY5-NRT1;1 affects P level in Arabidopsis thaliana? This is a vital question to improve the novelty of this story.
4. Related to Figure7 f, recommend to change "N-P alteration" to "P level".

Reviewer #3 (Remarks to the Author):

The concerns are properly addressed.

Reviewer #4 (Remarks to the Author):

This paper has been greatly improved by adding new data. However, I would like to recommend carefully rechecking the manuscript for minor errors. For example, "g-box" (line 161) should be "G-box," and "p-values" at the beginning of a sentence may be "p-Values." Another example is "MgCl₂" on line 576. A reference (Nat. Commun. 9: 1376) should be cited on line 135 because NIGT1.1 was defined in this paper.

REVIEWERS' COMMENTS

Reviewer #1 (Remarks to the Author):

My major criticisms in the previous version of the manuscript was related to the statistical analysis and in the present version of the manuscript, authors have included GXE analysis, which has indicated that the effects on N by temperature does not differ between the genotypes. This also raises the rationale of the focus on NRT1.1 and subsequent data analysis and whether the focus should be on ambient temperature N/P interactions or general effect, but that is something I will leave it to the authors. They have changed the wordings a bit, which I think is still a bit too strong and some areas. Authors are still arguing mostly their points, and present the story they like, perhaps more than what their data shows and I think they could make some attempts to make it a bit more clearer and data-driven. Nevertheless overall, it appears that the authors have addressed most of my comments and I do not have much further to add to the review.

Author response: We thank the reviewer for their comment and have made an effort to further make it clearer in the abstract and the discussion:

Line 22ff: “We also find that low external levels of these nutrients abolish root growth responses to high ambient temperature. We show that in Arabidopsis, this suppression is due to the function of the transcription factor ELONGATED HYPOCOTYL 5 (HY5) and its transcriptional regulation of the transceptor NITRATE TRANSPORTER 1.1 (NRT1.1).”

Line 407ff: “While the dependence of thermomorphogenesis on external nitrogen and phosphate levels is clearly governed by a HY5-NRT1.1 regulatory mechanism, only the effect of thermomorphogenesis on phosphate contents of plants seems to be dependent on the HY5-NRT1.1 mechanism. It is therefore not yet clear how and to which extent the thermomorphogenesis related changes of phosphate and nitrogen tissue levels and the modulation of thermomorphogenesis by the external levels of phosphate and nitrogen are related.”

Line 425 ff: “In a somehow surprising manner, the HY5-NRT1 regulatory module seems to affect plant tissue P-levels in response to higher temperature but not N-levels. This is even more surprising as we found that HY5 directly transcriptionally regulates NRT1 and other N-homeostasis genes, but we found no clear signature of a notable direct regulation of P-homeostatic genes.”

Reviewer #2 (Remarks to the Author):

The revised manuscript has greatly improved, but it still needs further improvement on the following points:

1. Line109-114, Thank you for measuring the nutrient contents in the 9-day old seedling roots. But the results show that the high ambient temperature related phenotypes may not dependent on the nutrient contents changing (eg. N, P levels) and HY5 mutation in 9-day old seedling roots. It is different from the responses of 4 week old plant shoots, which the phenotypes are related to the nutrient contents changing (eg. P level) and HY5 mutation. So it may indicate that there have two different regulatory mechanisms in 9 day old seedlings and 4 week old plants, or it exist two different regulatory mechanisms in the shoot and root under high ambient temperature treatments. Consider the nutrient contents data is a key results to link the high ambient temperature to HY5 and plant phenotypes, the results of nutrient contents in 9-day old seedling shoots and 4 week old plant roots are also needed. The difference of nutrient contents in 9-day old seedling and 4 week old plant could not easily explained by “nutrients accumulate over time”, more discussion is needed.

Author response: We thank the reviewer to draw our attention to this. To make this clear we modified line 111ff: “These data suggest that changes of nutrients accumulate over time in plants and then cumulatively affect nutrient contents in the shoot part of more mature plants. Alternatively, it is also possible that this is a time dependent regulatory process that starts later than 9 days after germination. Taken together our data show that the levels of N and P in plants are regulated in response to elevated temperature, and that in Arabidopsis HY5 is required for the regulation of temperature dependent P level changes.”

2. Related to figure 3b and c, whether NRT1;1 gene expression level also altered in 35S:HY5-GFP, 35S:HY5 S36A-GFP, and 35S:HY5 S36D-GFP lines under 21 /28 °C treatments.

Author response: We believe that our phenotypic data is very clear and additional experiments are not warranted at this time as the phenotypes are consistent with a recent study shows that phosphorylation of HY5 affects the binding affinity to the targets and thereby changes their expression (Wang et al., 2020). It is therefore likely that phosphorylation of HY5 affects the binding to NRT1.1 promoter and regulates expression and thus, explains the phenotype as seen in our study. Detailed experiments on how HY5 phosphorylation affects temperature will be performed in the future but are not in the scope of this study.

3. How HY5-NRT1;1 affects P level in Arabidopsis thaliana? This is a vital question to improve the novelty of this story.

Author response: We have clearly shown that HY5 and NRT1.1 affect P-levels in Arabidopsis. We have referred to previous work and about possible explanations:

Line 203ff: "NRT1.1 protein level is destabilized during P starvation in Arabidopsis (Medici et al., 2015) and the OsNRT1.1B-OsSPX4 module has been identified to integrate nitrate and phosphate signaling in rice (Hu et al., 2019)." We believe that further compressive molecular investigations are outside the scope of this manuscript as they will require significant future work.

4. Related to Figure 7 f, recommend to change "N-P alteration" to "P level".

Author response: We corrected it according to this comment.

Reviewer #3 (Remarks to the Author):

The concerns are properly addressed.

Author response: Thank you very much.

Reviewer #4 (Remarks to the Author):

This paper has been greatly improved by adding new data. However, I would like to recommend carefully rechecking the manuscript for minor errors. For example, "g-box" (line 161) should be "G-box," and "p-values" at the beginning of a sentence may be "p-Values." Another example is "MgCl2" on line 576. A reference (Nat. Commun. 9: 1376) should be cited on line 135 because NIGT1.1 was defined in this paper.

Author response: We thank the reviewer for pointing this out. Now we corrected these issues and checked the manuscript carefully.